# Tubulin cofactors and Arl2 are cage-like chaperones that regulate the soluble αβ-tubulin pool for microtubule dynamics

Stanley Nithianantham[1], Sinh Le[1], Elbert Seto[1], Weitao Jia[1], Julie Leary[1], Kevin D Corbett[2,3], Jeffrey K Moore[4], Jawdat Al-Bassam[1]*

[1]Department of Molecular Cellular Biology, University of California, Davis, Davis, United States; [2]Ludwig Institute for Cancer Research, University of California, San Diego, San Diego, United States; [3]Department of Cellular and Molecular Medicine, University of California, San Diego, San Diego, United States; [4]Department of Cell and Developmental Biology, University of Colorado School of Medicine, Aurora, United States

**Abstract** Microtubule dynamics and polarity stem from the polymerization of αβ-tubulin heterodimers. Five conserved tubulin cofactors/chaperones and the Arl2 GTPase regulate α- and β-tubulin assembly into heterodimers and maintain the soluble tubulin pool in the cytoplasm, but their physical mechanisms are unknown. Here, we reconstitute a core tubulin chaperone consisting of tubulin cofactors TBCD, TBCE, and Arl2, and reveal a cage-like structure for regulating αβ-tubulin. Biochemical assays and electron microscopy structures of multiple intermediates show the sequential binding of αβ-tubulin dimer followed by tubulin cofactor TBCC onto this chaperone, forming a ternary complex in which Arl2 GTP hydrolysis is activated to alter αβ-tubulin conformation. A GTP-state locked Arl2 mutant inhibits ternary complex dissociation in vitro and causes severe defects in microtubule dynamics in vivo. Our studies suggest a revised paradigm for tubulin cofactors and Arl2 functions as a catalytic chaperone that regulates soluble αβ-tubulin assembly and maintenance to support microtubule dynamics.

*For correspondence: jawdat@ ucdavis.edu

**Competing interests:** The authors declare that no competing interests exist.

## Introduction

Microtubules (MTs) are dynamic polymers that modulate fundamental cellular processes through dynamic αβ-tubulin polymerization and depolymerization at their ends, and serve as polarized tracks for molecular motor proteins (*Akhmanova and Steinmetz, 2008*). Polarity and dynamic instability are fundamental features of the MT polymer, originating from the head-to-tail polymerization of αβ-tubulin heterodimers (*Nogales et al., 1999*; *Alushin et al., 2014*). The αβ-tubulin dimer contains two GTP-binding sites: an inactive non-exchangeable site (N-site) on α-tubulin, which is suggested to stabilize αβ-tubulin dimers during their biogenesis, and an active exchangeable site (E-site) on β-tubulin, which is stimulated to hydrolyze GTP upon αβ-tubulin incorporation into MT lattices at the plus ends (*Nogales et al., 1999*; *Alushin et al., 2014*). GTP hydrolysis at the E-site leads to dynamic instability (catastrophe) at MT plus ends, due to the strain induced by the curvature of individual protofilaments (*Alushin et al., 2014*; *Brouhard and Rice, 2014*). Intracellular MT dynamics critically relies on a tightly controlled pool of soluble αβ-tubulin dimers in the cytoplasm. Despite their importance, the mechanisms for biogenesis, maintenance, and degradation of soluble αβ-tubulin dimers remain poorly understood (*Tian and Cowan, 2013*).

αβ-tubulin is maintained at a high concentration (∼6 µM) in the cytoplasm through regulation of translation from tubulin mRNAs (*Cleveland et al., 1978*; *Cleveland, 1989*). α- and β-tubulin are

**eLife digest** Cells contain a network of protein filaments called microtubules. These filaments are involved in many biological processes; for example, they help cells keep the right shape, and they help to transport proteins and other materials inside cells.

Two proteins called α-tubulin and β-tubulin are the building blocks of microtubules. The filaments are very dynamic structures that can rapidly change length as individual tubulin units are either added or removed to the filament ends. Several proteins known as tubulin cofactors and an enzyme called Arl2 help to build a vast pool of tubulin units that are able attach to the microtubules. These units—called αβ-tubulin—are formed by α-tubulin and β-tubulin binding to each other, but it not clear exactly what roles the tubulin cofactors and Arl2 play in this process.

Nithianantham et al. used a combination of microscopy and biochemical techniques to study how the tubulin cofactors and Arl2 are organised, and their role in the assembly of microtubules in yeast. The experiments show that Arl2 and two tubulin cofactors associate with each other to form a stable 'complex' that has a cage-like structure. A molecule of αβ-tubulin binds to the complex, followed by another cofactor called TBCC. This activates the enzyme activity of Arl2, which releases the energy needed to alter the shape of the αβ-tubulin. Nithianantham et al. also found that yeast cells with a mutant form of Arl2 that lacked enzyme activity had problems forming microtubules.

Together, these findings show that the tubulin cofactors and Arl2 form a complex that regulates the assembly and maintenance of αβ-tubulin. The next challenge is to understand how this regulation influences the way that microtubules grow and shrink inside cells.

translated and folded as monomers in the type II chaperonin TRIC/CCT (*Lewis et al., 1997*). Biogenesis and degradation of the αβ-tubulin heterodimer are non-spontaneous processes that rely on five highly conserved tubulin cofactor (TBC) proteins: TBCA, TBCB, TBCC, TBCD, and TBCE (described in *Figure 1A*; *Lewis et al., 1997*; *Lundin et al., 2010*). Orthologs of these proteins have been identified in all eukaryotes studied to date (*Lewis et al., 1997*; *Lundin et al., 2010*). The maintenance of a concentrated pool of tubulin dimers by the TBC proteins is essential for proper MT dynamics in eukaryotic cells (*Tian et al., 1996*; *Lewis et al., 1997*; *Lundin et al., 2010*). The TBC proteins' functions are finely balanced: their loss or their overexpression are both lethal in most eukaryotes, stemming from a complete loss of the MT cytoskeleton (*Steinborn et al., 2002*; *Lacefield et al., 2006*; *Jin et al., 2009*). In budding yeast, the first identified chromosomal instability (CIN) phenotypes, showing severe mitotic spindle defects due to loss of MTs, were ultimately traced to loss of TBC proteins (*Hoyt et al., 1990*, *1997*; *Antoshechkin and Han, 2002*; *Steinborn et al., 2002*; *Lacefield et al., 2006*; *Jin et al., 2009*). In humans, missense mutations in TBCE and TBCB are linked to hypo-parathyroidism facial dysmorphism (also termed Kenny-Caffey syndrome) and giant axonal neuropathy, in which developmental defects are observed due to impairment of MT cytoskeleton function (*Parvari et al., 2002*; *Wang et al., 2005*). In addition to the five conserved TBC proteins, the small Arl2 GTPase (<u>A</u>DP <u>R</u>ibosylation <u>F</u>actor-<u>L</u>ike-2) regulates the function of TBC proteins in αβ-tubulin biogenesis/degradation through an unknown mechanism (*Figure 1A*). Although Arl2 is not considered a tubulin cofactor, its loss causes nearly identical defects to those observed with TBCC, TBCD, or TBCE loss (*Hoyt et al., 1997*; *Radcliffe et al., 2000*; *Mori and Toda, 2013*).

A stepwise αβ-tubulin biogenesis/degradation paradigm has been proposed based on genetic and biochemical studies (*Tian et al., 1996*; *Lewis et al., 1997*; *Lundin et al., 2010*; shown in *Figure 1B*), in which TBC proteins form dynamic assemblies to dimerize αβ-tubulin, as follows: (1) TBCA and TBCB bind β-tubulin and α-tubulin monomers, respectively, after their folding; (2) TBCA hands off β-tubulin to TBCD, and TBCB hands off α-tubulin to TBCE; (3) TBCC drives association of TBCD and TBCE with their bound α- and β-tubulin monomers, to form a 'super-complex' that forms and activates the αβ-tubulin dimer (*Tian and Cowan, 2013*); and (4) Arl2 is simulated to hydrolyze GTP through the GTPase activating protein (GAP) function of TBCC. The role of Arl2 GTP hydrolysis in this pathway remains unknown (*Bhamidipati et al., 2000*); Arl2 and its activation by TBCC have been suggested to operate in parallel to the TBC pathway (*Figure 1B*). However, the roles for TBCC and the Arl2 GTPase remain poorly understood (*Tian et al., 1999*; *Mori and Toda, 2013*). Overexpression of TBC proteins results in one of two unique phenotypes: TBCA or TBCB overexpression in budding or fission yeast

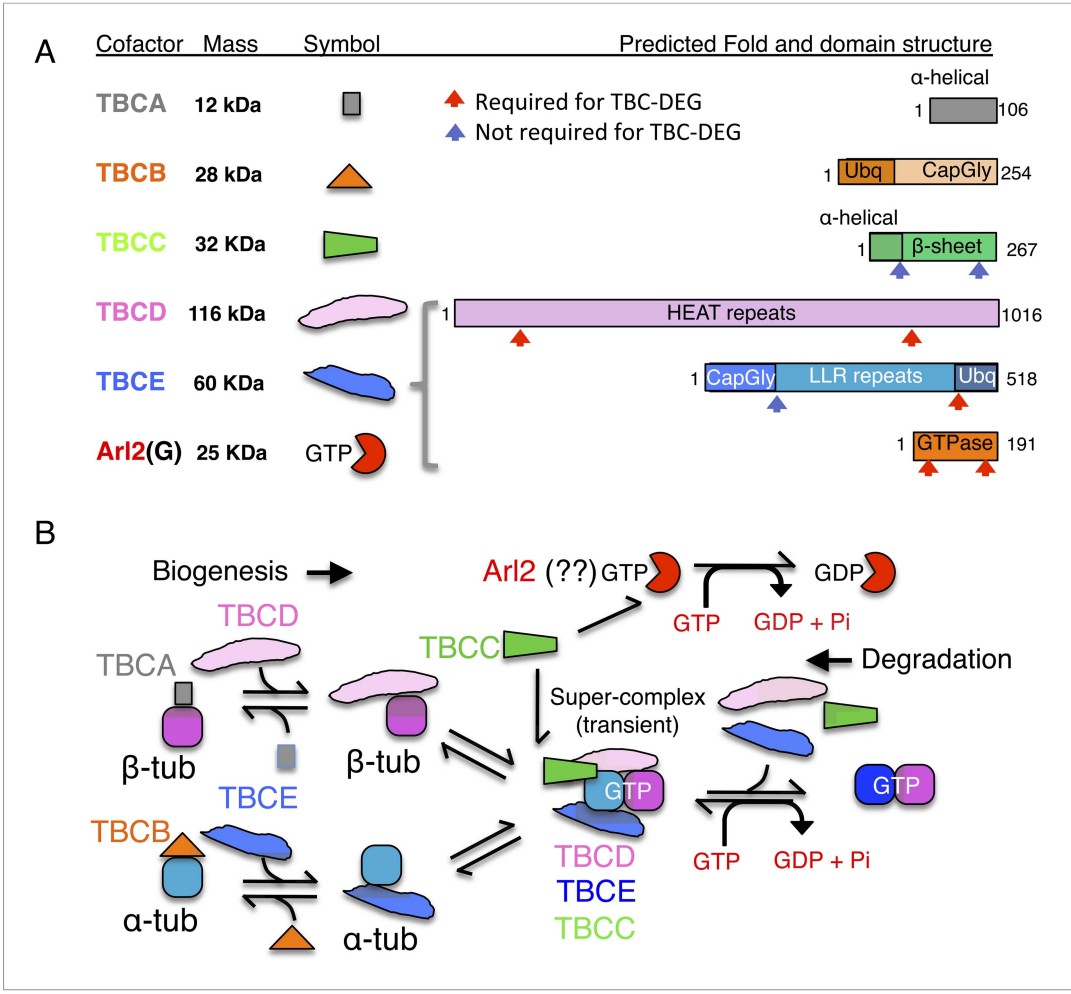

**Figure 1**. Tubulin cofactors and Arl2 GTPase: domain organization and paradigm for function. (**A**) Tubulin cofactors A–E, Arl2 GTPase masses, and domain organization. TBCA and TBCB co-expression is not required for TBC-DEG expression. Red arrowheads mark domains required for forming TBC-DEG complex assembly. Blue arrowheads mark domains not required for TBC-DEG complex assembly. (**B**) Initial paradigm for tubulin cofactors and Arl2 activities based on previous studies. Each of the molecules is suggested to be monomeric, and only assemble into complexes to drive αβ-tubulin biogenesis or degradation, via interactions regulated by dynamic equilibria. TBCA binds nascent β-tubulin and TBCB binds nascent α-tubulin. TBCA and TBCB are replaced by TBCD and TBCE, respectively. TBCC drives TBCE-α-tubulin and TBCD-β-tubulin to form a supercomplex. GTP hydrolysis in Arl2 is activated by TBCC in a parallel pathway to tubulin assembly. Tubulin biogenesis and degradation intermediate bind and form tubulin dimers, a process that requires Arl2 and tubulin to undergo GTP hydrolysis as an energy source. (Adopted from *Lewis et al., 1997*.)

suppresses defects induced by overexpression of α- or β- tubulin, but does not otherwise affect MT dynamics. In contrast, overexpression of TBCC, TBCD, TBCE, or Arl2 leads to rapid MT loss (*Archer et al., 1998*; *Feierbach et al., 1999*; *Radcliffe et al., 1999*; *Lacefield et al., 2006*).

Here, we show that TBCD, TBCE, and Arl2 assemble into a stable heterotrimeric chaperone (TBC-DEG) with a cage-like structure. This chaperone binds αβ-tubulin and TBCC sequentially, serving as a catalytic platform powered by the Arl2 GTPase for αβ-tubulin assembly and activation. A soluble αβ-tubulin dimer binds TBC-DEG and primes Arl2, followed by TBCC binding and GTP hydrolysis activation. We show that TBCC is a unique GAP for which affinity depends on αβ-tubulin binding onto TBC-DEG. TBCC promotes GTP hydrolysis through its C-terminal β-helix domain, which interfaces with both Arl2 and αβ-tubulin in a ternary complex. We further find that in *Saccharomyces cerevisiae* cells, a mutation locking the Arl2 GTPase into a GTP-bound state profoundly affects MT dynamics.

Overall, our studies reveal a new role for tubulin cofactors TBCD, TBCE, and Arl2, which together assemble a GTP-hydrolyzing tubulin chaperone critical for the biogenesis, maintenance, and degradation of soluble αβ-tubulin, defects in which have a profound effect on MT dynamics in vivo. The finding that αβ-tubulin is assembled on a multi-subunit platform establishes a new paradigm for the mechanisms of the TBC proteins in tubulin biogenesis, maintenance, and degradation (*Figure 1B*).

## Results

### Tubulin cofactors TBCD, TBCE, and the Arl2 GTPase form a stable heterotrimeric chaperone

To gain insight into the molecular mechanisms of tubulin cofactors and Arl2, we expressed the *S. cerevisiae* orthologs of TBCA, TBCB, TBCC, TBCD, TBCE, and Arl2 (named Rbl2, Alf1, Cin1p, Pac2p, Cin2p, and Cin4p, and referred to hereafter as TBCA, TBCB, TBCC, TBCD, TBCE, and Arl2 [*Figure 1A*]) both individually and in combinations, with the goal of reconstituting relevant complexes. TBCA and TBCB are small proteins (12 and 28 kDa in *S. cerevisiae*) that have been suggested to sequester monomeric β- and α-tubulin, respectively, while TBCC, TBCD, TBCE, and Arl2 regulate αβ-tubulin dimer biogenesis and degradation through unknown mechanisms (*Archer et al., 1998*; *Feierbach et al., 1999*; *Lundin et al., 2010*). Sequence alignments and structure predictions identify conserved domains within each protein (*Figure 1A*), but the molecular functions of these domains remain unknown. We found that TBCA, TBCB, and TBCC are each soluble when expressed on their own in *Escherichia coli*, while TBCD, TBCE, and Arl2 are insoluble on their own (see 'Materials and methods'). Co-expression of these three proteins, however, results in a stable and homogenous TBCD-TBCE-Arl2 GTPase complex that we term TBC-DEG (*Figure 2A*). When we coexpressed TBCA, TBCB, or TBCC with TBC-DEG, we observed no interaction with TBCA or TBCB, and an unstable, transient interaction with TBCC (as determined by mass spectrometry; *Table 1*). Size exclusion chromatography with multi-angle light scattering (SEC-MALS) demonstrates that TBC-DEG is a 205 kDa heterotrimer with one copy of each protein (*Figure 2A,C,D*; *Table 2*). Similar analysis shows that TBCC is a 32 kDa monomer, and porcine brain αβ-tubulin is a 100 kDa heterodimer, as shown previously (*Figure 2A,C,D*; *Table 2*). Monomeric TBCD, TBCE, or Arl2 subunits were not observed in vitro at any concentration and the TBC-DEG complex behaves as a single biochemical entity (*Figure 2A,C,D*; *Table 2*). At high ionic strength, TBC-DEG complexes precipitate, presumably due to dissociation and insolubility of individual subunits (data not shown). A recent study suggests human TBCE is soluble and forms complexes with TBCB (*Serna et al., 2015*). We do not observe TBCE-TBCB complexes using our TBC protein bacterial expression system. TBCE is insoluble without TBCD and Arl2 coexpression in bacteria. We believe that TBCE solubility maybe due to its expression in a eukaryotic system, where assembly and co-purification with TBCD and Arl2 is possible.

To understand the role of conserved domains within the TBC-DEG complex, we systematically deleted predicted domains in each subunit (see 'Materials and methods'; *Figure 2—figure supplement 1A,B*). Deletion of either the N- or C-terminal domains of both TBCD and Arl2, or the C-terminal ubiquitin-like domain of TBCE, leads to insoluble TBC-DEG that cannot be purified from *E. coli*. In contrast, deleting the N-terminal Cap-Gly domain of TBCE, predicted to bind the C-terminal tail of α-tubulin, did not affect assembly of soluble TBC-DEG complexes. We next determined the effect of inserting small (6xHis) or large (GFP, green fluorescent protein) tags on TBC-DEG assembly. Consistent with the deletion analysis, large or small tags were not tolerated on either end of Arl2 or at the C-termini of TBCD or TBCE (*Figure 2—figure supplement 1A,B*). Both 6xHis and GFP tags were tolerated at the N-termini of TBCD and TBCE (*Figure 2—figure supplement 1A,B*). These data suggest that the conserved domains of TBCD, TBCE, and Arl2 are required to assemble a TBC-DEG complex, in which the N-termini of TBCD and TBCE are exposed, while both termini of Arl2, and the C-termini of TBCD and TBCE, are buried and do not tolerate insertions.

### αβ-tubulin and TBCC sequentially bind TBC-DEG depending on the state of the Arl2 GTPase

We next sought to test the idea that TBC-DEG serves as a platform for soluble αβ-tubulin dimer assembly, and to examine the role of the Arl2 GTPase in this assembly. TBC-DEG binds soluble αβ-tubulin dimers with high affinity, forming stable complexes with a measured mass of 308 kDa (*Figure 2A,C,D*; *Table 2*), indicating that a single TBC-DEG (200 kDa) binds a single αβ-tubulin dimer

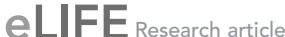

**Figure 2**. Hierarchical assembly of TBCC with TBC-DEG and soluble αβ-tubulin dimer binding in the GDP·Pi state. (**A**) Size exclusion chromatography (SEC) intensity traces of TBC-DEG (black), TBC-DEG:αβ-tubulin (cyan), αβ-tubulin (red), and TBCC (purple). (**B**) SEC intensity traces of TBC-DEG+TBCC+αβ-tubulin-GDP·ALF$_x$ (green), TBC-DEG+TBCC+αβ-tubulin-GTP (gray), TBC-DEG+TBCC-GTP-ALF$_x$ (black), TBCC+αβ-tubulin (blue), and αβ-tubulin+TBCC (blue). Additional states are described in *Figure 2—figure supplement 1C,D*. (**C**) Composition of SEC fractions shown in **A** and **B** using SDS-PAGE. Panel I, TBC-DEG; panel II, TBC-DEG:αβ-tubulin; panel III, TBC-DEG+TBCC-GDP·ALF$_x$; panel IV, TBCC+αβ-tubulin; panel V, TBC-DEG+TBCC+αβ-tubulin-GTP; and panel VI, TBC-DEG+TBCC+αβ-tubulin-GDP·ALF$_x$. TBC-DEG forms an active heterotrimeric complex, and TBCC forms a complex that

*Figure 2. continued on next page*

*Figure 2. Continued*

co-migrates with TBC-DEG upon αβ-tubulin binding in the presence of GDP·ALF$_x$ (panel IV). The protein standard is shown on the left and proteins are marked on the right. TBC-DEG complexes interact weakly with the resin media leading to wide elution SEC profiles in most conditions. (**D**) Molecular masses of TBC-DEG, αβ-tubulin, TBCC, and their complexes measured using size exclusion chromatography with multi-angle light scattering (SEC-MALS). Solid lines represent SEC intensity traces on an intensity scale shown on the right y-axis, and dotted lines represent masses calculated on the mass scale shown on the left y-axis; TBC-DEG (black), αβ-tubulin (red), TBCC (purple), TBC-DEG:αβ-tubulin (cyan), and TBC-DEG:αβ-tubulin:TBCC-GDP·ALF$_x$ (green). Masses and elution volumes are detailed in *Table 2*. (**E**) Scheme for the hierarchical assembly of TBC-DEG with TBCC and αβ-tubulin and the role of nucleotide. TBCD, TBCE, and Arl2 form TBC-DEG complexes (TBC-DEG) and bind a single αβ-tubulin dimer (αβ-tub) to form TBC-DEG:αβ-tubulin (TBC-DEG:αβ-tub), which recruits TBCC in the GTP-like state to form TBC-DEG:αβ-tubulin:TBCC (TBC-DEG:αβ-tub:TBCC).

The following figure supplement is available for figure 2:

**Figure supplement 1**. Tubulin cofactor-Arl2 co-expression and biochemical studies on TBC-DEG constructs.

(110 kDa). The TBC-DEG:αβ-tubulin complex is likely to be the ~300 kDa tubulin biogenesis intermediate identified two decades ago by *Paciucci (1994)*. We next determined the conditions for TBCC binding to TBC-DEG and αβ-tubulin. TBCC does not bind either αβ-tubulin or TBC-DEG in isolation, but strongly interacts with the TBC-DEG:αβ-tubulin complex (*Figure 2A–C*). This interaction strongly depends on the GTP nucleotide present during complex assembly. We observed TBCC binding to the TBC-DEG:αβ-tubulin complex when incubated with the non-hydrolysable GTP analog GTPγS or the transition state analog GDP·ALF$_x$, but no binding in the presence of GTP or GDP (*Figure 2B–C*, *Figure 2—figure supplement 1 B–E*). We measured a 340 kDa mass for the TBC-DEG:αβ-tubulin:TBCC ternary complex, indicating that the TBC-DEG:αβ-tubulin complex (310 kDa) associates with a single molecule of TBCC (34 kDa) (*Table 2*; *Figure 2B–D*). To determine if the Arl2 GTPase is responsible for increasing the TBCC binding affinity to TBC-DEG, we next generated a Gln73Leu (Q73L) mutation in Arl2, which inhibits GTP hydrolysis and results in a 'GTP-locked' state (*Veltel et al., 2008*). Bacterial expression of recombinant Arl2-Q73L shows that it assembles with TBCD and TBCE into a TBC-DEG-Q73L complex. In contrast to TBC-DEG, TBC-DEG-Q73L interacts with TBCC in the absence of αβ-tubulin (*Figure 3A,B*, panel I), and assembles with αβ-tubulin and TBCC to form a stable and fully saturated ternary complex (*Figure 3A,B*, panel II; mass of 335 kDa by SEC-MALS; *Table 2*) in the presence of GTP. Thus, our biochemical reconstitutions indicate that TBCC binding to TBC-DEG is promoted by both αβ-tubulin binding to TBC-DEG and the GTP-bound state of Arl2 (*Figure 2D*). These findings support a proposed role for TBCC as a GAP for Arl2, whose association with the TBC-DEG chaperone is responsive to αβ-tubulin binding.

## Sequential binding of αβ-tubulin and TBCC activates maximal GTP hydrolysis in TBC-DEG

Next, we studied the GTP hydrolysis activity of TBC-DEG and the effect of αβ-tubulin and TBCC binding, using a free-phosphate detection assay (*Figure 3C*; *Table 3*). In the absence of other factors, TBC-DEG hydrolyzes GTP extremely slowly (*Figure 3C*; *Table 3*). Addition of equimolar αβ-tubulin, which alone does not show detectable GTP hydrolysis in this assay, stimulates a modest level of GTP hydrolysis activity in TBC-DEG ($k_{cat}$ = ~0.40 min$^{-1}$; *Figure 3D*, *Figure 3—figure supplement 1C*; *Table 3*). In contrast, the addition of equimolar TBCC to TBC-DEG activates a substantially higher rate of GTP hydrolysis ($k_{cat}$ = ~0.77 min$^{-1}$), consistent with its proposed function as a GAP for Arl2 (*Tian et al., 1997*; *Bhamidipati et al., 2000*; *Mori and Toda, 2013*; *Newman et al., 2014*). When equimolar amounts of both αβ-tubulin and TBCC are added to TBC-DEG, GTP hydrolysis was stimulated twofold more than in the presence of TBCC alone ($k_{cat}$ = 1.85 min$^{-1}$; *Figure 3—figure supplement 1A*). This increase may be due either to an increase in the affinity of TBCC for Arl2 in the presence of αβ-tubulin, or activation of GTP hydrolysis within the bound αβ-tubulin itself. To distinguish these models, we next assayed GTP hydrolysis of TBC-DEG-Q73L in the presence of equimolar αβ-tubulin and TBCC. This complex shows low GTP hydrolysis activity ($k_{cat}$ = 0.5 min$^{-1}$) with a high $K_m$ (387 μM), supporting the idea that within the ternary complex, αβ-tubulin contributes only a small fraction of the total GTP hydrolysis activity. Taken together, our data provide a new context to explain extensive prior genetic and biochemical data on the role of Arl2 in regulating tubulin cofactor activity (*Tian et al., 1997*;

**Table 1.** Identification of tubulin cofactor subunits* using nano-LC-MS/MS

| Protein name | Molecular mass | pI | Peptide coverage |
|---|---|---|---|
| Yeast TBCD (cin1p) | 116,647.8 Da | 8.53 | 82.1% |
| Yeast TBCE (pac2p) | 59,257.6 Da | 8.77 | 79.3% |
| Yeast Arl2 (cin4p) | 22,066.6 Da | 5.70 | 85.9% |
| Yeast TBCC (cin2p) | 34,045.4 Da | 7.05 | 9.0% |

*Identified from His-TBCD TBCE, TBCC, TBCB, TBCA, and Arl2 co-expression.

*Bhamidipati et al., 2000*; *Mori and Toda, 2013*; *Newman et al., 2014*). Our studies reveal that TBCC is a novel αβ-tubulin-responsive GAP that activates Arl2 in the context of the TBC-DEG chaperone.

Next, we explored how the TBCC and αβ-tubulin concentration influences GTP hydrolysis by TBC-DEG. We measured the steady-state GTP hydrolysis of TBC-DEG titrated with a range of TBCC and αβ-tubulin concentrations. Decreasing the molar ratio of TBCC to TBC-DEG, while maintaining a stoichiometric amount of αβ-tubulin, increases the apparent $K_m$ for GTP hydrolysis, further supporting the idea that TBCC is a true GAP (*Figure 3F*, *Figure 3—figure supplement 1H*; *Table 3*; *Veltel et al., 2008*). In contrast, increasing the ratio of αβ-tubulin to TBC-DEG (0–3 μM), while maintaining a stoichiometric amount of TBCC, stimulates a step-wise increase in the maximal rate of GTP hydrolysis (*Figure 3E*; *Table 3*; Figure 3—figure supplement 3I). At 3 μM αβ-tubulin and at a 1:3:1 ratio of TBCC:αβ-tubulin:TBC-DEG, we observe the highest GTP hydrolysis rate (*Table 3*: $k_{cat}$ = 3.0 min⁻¹), suggesting that each TBC-DEG chaperone can undergo multiple rounds of GTP hydrolysis, upon binding and releasing multiple αβ-tubulin dimers during each experiment (*Figure 3E*). We were unable to test αβ-tubulin concentrations higher than 3 μM in this assay, as at 6 μM αβ-tubulin or higher we expect αβ-tubulin polymerization into MTs to significantly contribute to the overall GTP hydrolysis observed. Within the tested αβ-tubulin concentration range (0–3 μM), the TBC-DEG GTP hydrolysis rate ($k_{cat}$) is proportional to the αβ-tubulin concentration, starting at ~0.8 min⁻¹ in the absence of αβ-tubulin (*Figure 3E*, Figure 3—figure supplement 3H) and climbing and then plateauing at 3.0 min⁻¹ at 3 μM αβ-tubulin. Thus, TBCC is an αβ-tubulin dependent GAP that activates TBC-DEG GTP hydrolysis in a cyclic manner where the degree of GAP activity depends on the soluble αβ-tubulin concentration.

## The TBC-DEG complex is a cage-like chaperone with a hollow central core

To determine the 3D structure of the TBC-DEG-Q73L chaperone, we used electron microscopy (EM) and single-particle image analysis. While cryo-EM imaging of TBC-DEG was not possible due to solubility defects and aggregation in vitreous ice, we were able to collect negative-stain EM data and

**Table 2**. Size exclusion chromatography (SEC) and SEC with multi-angle light scattering (SEC-MALS) parameters for tubulin cofactor and αβ-tubulin complexes

| Protein complex | Elution volume | Predicted | Apparent | SEC-MAL | Stokes $R$ |
|---|---|---|---|---|---|
| TBC-DEG | 11.5 ml | 198 kDa | 213 kDa | 215 ± 10 kDa | ~50 Å |
| TBC-DEG:αβ-tub | 10.7 ml | 308 kDa | 322 kDa | 310 ± 10 kDa | ~56 Å |
| TBC-DEG:C:αβ-tub-GDP.ALF$_x$ | 10.2 ml | 342 kDa | 376 kDa | 335 ± 10 kDa | ~59 Å |
| TBC-DEG-Q73L | 10.6 ml | 198 kDa | 218 kDa | N/A | ~50 Å |
| TBC-DEG-Q73L:αβ-tub | 10.7 ml | 308 kDa | 322 kDa | N/A | ~56 Å |
| TBC-DEGQ73L:C:αβ-tub | 10.2 ml | 342 kDa | 376 kDa | 340 ± 10 kDa | ~59 Å |
| αβ-tubulin dimer | 12.9 ml | 100 kDa | 103 kDa | 100 ± 5 kDa | ~39 Å |
| TBCC | 15.0 ml | 34 kDa | 35 kDa | 30 ± 5 kDa | ~27 Å |

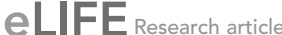

**Figure 3**. TBCC activates dual GTP hydrolyses in Arl2 and αβ-tubulin on TBC-DEG: αβ-tubulin complexes. (**A**) Size exclusion chromatography (SEC) intensity traces of TBC-DEG-Arl2-Q73L (TBC-DEG-Q73L) assembly with TBCC and αβ-tubulin; TBC-DEG-Q73L+TBCC (black), TBC-DEG-Q73L+αβ-tubulin+TBCC (green), αβ-tubulin (red), and TBCC (purple). (**B**) Analysis of SEC fractions described in **A** by SDS-PAGE. Panel I, TBC-DEG-Q73L+TBCC+GTP; panel II, TBC-DEG-Q73L+TBCC+αβ-tubulin-GTP. (**C**) Scheme for GTP hydrolysis by TBC-DEG and the effect of αβ-tubulin binding and TBCC on the GTP hydrolysis pathway. (**D**) Steady-state GTP hydrolysis assays of different 1 μM TBC-DEG, αβ-tubulin, and TBCC assemblies. TBC-DEG (red) and TBC-DEG+αβ-tubulin (orange) hydrolyze GTP very slowly. TBCC+αβ-tubulin (black) hydrolyzes negligible amounts of GTP. TBC-DEG+αβ-tubulin+TBCC hydrolyzes GTP (blue; 1.8 min⁻¹) at a rate roughly twofold higher than TBC-DEG+TBCC (green; 0.8 min⁻¹). $K_m$ and $k_{cat}$ values are reported in *Table 3*. (**E**) The effect of αβ-tubulin binding on TBC-DEG GTP hydrolysis. Top panel, scheme for GTP hydrolysis by TBC-DEG and the effect of limiting or varying the αβ-tubulin concentration on GTP hydrolysis. Bottom

*Figure 3. Continued*

panel, titrating αβ-tubulin concentrations (0–3.0 µM) to 1 µM TBC-DEG and 1 µM TBCC. The curves are labeled with the concentration at the plateau point for each curve. (**F**) The effect of TBCC concentration on TBC-DEG GTP hydrolysis. Top panel, scheme for GTP hydrolysis by TBC-DEG and the effect of limiting or varying the TBCC concentration on GTP hydrolysis. Bottom panel, titrating TBCC concentration (0.12–1.0 µM) to 1 µM TBC-DEG and 1 µM αβ-tubulin. The curves are labeled with the concentration at the plateau point for each curve. (**G**) The effect of Arl2-Q73L on TBC-DEG GTP hydrolysis. Top panel, scheme for GTP hydrolysis by TBC-DEG-Q73L and the effect of αβ-tubulin binding and TBCC on the GTP hydrolysis reaction. Bottom panel, steady-state GTP hydrolysis assays of 1 µM TBC-DEG+αβ-tubulin+TBCC (blue) compared to TBC-DEG-Q73L+αβ-tubulin+TBCC (purple). $K_m$ and $k_{cat}$ values are reported in *Table 3*.

The following figure supplement is available for figure 3:

**Figure supplement 1**. Summary of the catalytic GTP hydrolysis rates for different reconstitutions of TBC-DEG with TBCC and αβ-tubulin and the effects of Arl2 and TBCC mutations.

generate a robust medium-resolution 3D reconstruction. We collected a total of 20,000 particle images from 160 initial wide-field images, which showed a globular ~100 × 100 × 100 Å particle with significant internal structure (*Figure 4A*, *Figure 4—figure supplement 1A*). Particle orientations were well distributed, and reference-free classification showed homogeneous class averages representing a large range of views (*Figure 4—figure supplement 1B*). We generated a starting model using a common-lines approach based on prominent classes and then used projection matching and angular reconstitution, and refined a 3D map to 24 Å resolution (see 'Materials and methods'; *Figure 4A*,

**Table 3**. Steady-state GTP hydrolysis parameters for TBC-DEG, TBCC, and αβ-tubulin

| GTP hydrolysis reactions | $K_m$ (GTP) | $k_{cat}$ (min$^{-1}$/µM) |
|---|---|---|
| 1 µM TBC-DEG | 400 ± 30 µM | 0.06 ± 0.01 |
| 1 µM TBCC:1 µM αβ-tub | 20 ± 15 µM | 0.00 ± 0.01 |
| 1 µM TBC-DEG:1 µM αβ-tub | 69 ± 12 µM | 0.40 ± 0.01 |
| 1 µM TBC-DEG:1 µM TBCC | 94 ± 10 µM | 0.77 ± 0.04 |
| 1 µM TBC-DEG:1 µM TBCC:1 µM αβ-tub | 99 ± 10 µM | 1.88 ± 0.03 |
| 1 µM TBC-DEG-Q73L:1 µM TBCC:1 µM αβ-tub | 371 ± 20 µM | 0.50 ± 0.05 |
| 1 µM TBC-DEG:1 µM TBCC:0.25 µM αβ-tub | 87 ± 5 µM | 1.10 ± 0.06 |
| 1 µM TBC-DEG:1 µM TBCC:0.5 µM αβ-tub | 112 ± 4 µM | 1.41 ± 0.05 |
| 1 µM TBC-DEG:1 µM TBCC:1.5 µM αβ-tub | 98 ± 8 µM | 2.38 ± 0.08 |
| 1 µM TBC-DEG:1 µM TBCC:3.0 µM αβ-tub | 88 ± 9 µM | 2.77 ± 0.11 |
| 1 µM TBC-DEG:0.12 µM TBCC:1 µM αβ-tub | 149 ± 7 µM | 1.53 ± 0.04 |
| 1 µM TBC-DEG:0.25 µM TBCC:1 µM αβ-tub | 129 ± 3 µM | 1.69 ± 0.02 |
| 1 µM TBC-DEG:0.50 µM TBCC:1 µM αβ-tub | 108 ± 5 µM | 1.74 ± 0.04 |
| 1 µM TBC-DEG:1 µM TBCC-R186A:1 µM αβ-tub | 40 ± 10 µM | 0.54 ± 0.04 |
| 1 µM TBC-DEG:1 µM TBCC-Δ233-245:1 µM αβ-tub | 35 ± 8 µM | 0.34 ± 0.03 |
| 1 µM TBC-DEG:1 µM TBCC-Cterm:1 µM αβ-tub | 56 ± 7 µM | 1.13 ± 0.07 |

**Figure 4**. TBC-DEG complexes are compact cage-like chaperone assemblies with hollow cores. (**A**) Left panel, an expanded negative-stain image of TBC-DEG Q73L showing the cage-like assemblies. Middle panel, higher magnification view of the TBC-DEG-Q73L. Right panel, reference-free class averages (from *Figure 4—figure supplement 1C*) of TBC-DEG Q73L showing the variety of views. (**B**) A refined 24 Å TBC-DEG-Q73L 3D map shown in three rotated views. The floor, bow, trunk, pillar, and thumb regions are marked in each view. (**C**) Segmented 24 Å

*Figure 4. continued on next page*

*Figure 4. Continued*

TBC-DEG map with all unique segmented domains based on tagging assignment of the TBCE N-termini (*Figure 4—figure supplement 2B*). The bow region (blue) includes two globular ends: the ubiquitin domain (cyan) and the Cap-Gly domain (deep blue). Three interfaces stabilize the TBC-DEG cage: the bow pillar, the pillar floor, and the bow floor via the trunk. *Video 1* shows the **A** and **B** views. (**D**) A TBC-DEG subunit domain map shown to length scale. TBCD (pink, top panel) is predicted to consist of HEAT repeats. TBCE (middle panel) consists of an N-terminal Cap-Gly domain (dark blue), a leucine rich repeat (LRR) domain (blue), and a C-terminal ubiquitin-like domain (cyan). Arl2 (bottom panel) consists of a GTPase fold (orange). Colors correspond to subunits shown in **D**–**I**. (**E**) Pseudo-atomic TBC-DEG cage model showing the TBCD, TBCE, and Arl2 domain organization in assembling the cage structure. Each 3D map region is shown in a glossy color, and x-ray structures for orthologs fitted are shown as ribbons in the same color. The floor and thumb segments (pink) were fitted by the Cse1 crystal structure. The bow segment (blue) was fitted by the TLR4 LRR domain x-ray structure. The pillar segment (orange) was fitted by the x-ray structure of Arl2 (orange). The positions N-GFP-TBCE (dark green) and N-GFP-TBCD (light green) are shown. The trunk region (purple) was not fit with any atomic model. (**F**) A 90˚ vertically rotated view of that shown in **D**. (**G**) A 90˚ horizontally rotated view of that shown in **D**. (**H**) A central slice view of a 90˚ counterclockwise horizontally rotated view of of that shown in **D**. *Video 1* shows the **C**–**F** views. (**I**) Cartoon view of TBC-DEG domain organization comparable to the view shown in **F**. (**J**) Cartoon view of TBC-DEG domain organization comparable to the view shown in **G**.

The following figure supplements are available for figure 4:

**Figure supplement 1**. Electron microscopy and 3D reconstruction of the TBC-DEG-Q73L cage-like chaperones.

**Figure supplement 2**. Mapping the TBCE N-terminal Cap-Gly domain using TBC-DE(N-GFP) fusion and 3D reconstructions.

**Figure supplement 3**. Cumulative docking of atomic models for TBCD, TBCE paralogs, and Arl2 without segmentation using low-resolution model filtering.

*Figure 4—figure supplement 1C*, *Table 4*). 2D projections generated from the refined 3D map matched well to the reference-free class averages (*Figure 4—figure supplement 1D*). We obtained matching reconstructions using a variety of low-resolution starting models, which converged during angular refinement to the 3D reconstructions described below.

The 3D reconstruction of TBC-DEG-Q73L shows a compact cage-like structure with a hollow core, and overall dimensions of 120 × 100 × 90 Å (*Figure 4B*). TBC-DEG consists of a circular 'floor' with a large vertical 'thumb' extension (*Figure 4B*, pink), facing a 'bow' density with two globular ends (*Figure 4B*, blue). The bow (*Figure 4B*, blue) is attached at its center to the floor (*Figure 4B*, pink) via a 'trunk' density and binds a 'pillar' density (*Figure 4B*, orange) via one of its globular ends (*Figure 4B*, cyan). Three interfaces form the TBC-DEG cage structure: (1) bow to floor interface via the trunk; (2) bow to pillar interface (cyan); and (3) floor to pillar interface. To determine the locations of individual TBC-DEG subunits within this structure, we imaged a complex with N-terminally GFP-fused TBCE (TBC-DE(N-GFP)G) using negative-stain EM (*Figure 2—figure supplement 1G*). TBC-DE(N-GFP)G class averages show the addition of ordered density when compared to equivalent class averages of TBC-DEG (*Figure 4—figure supplement 2A*). We determined a 24 Å 3D structure for TBC-DE (N-GFP)G using a 50 Å resolution filtered TBC-DEG as a starting model (*Figure 4—figure*

**Table 4**. Electron microscopy Fourier shell correlation (FSC) resolution analyses

| Complex | Particle images | Resolution (Å)* |
| --- | --- | --- |
| TBC-DEG-Q73L | 16,000 | 24.0 |
| TBC-DEG-Q73L-αβ-tub | 19,000 | 24.0 |
| TBC-DEG-Q73L-αβ-tub:TBCC | 18,000 | 24.0 |
| TBC-DE(N-GFP)G | 15,000 | 24.0 |

*Resolution cross-correlation criterion cut-off set at 0.5.

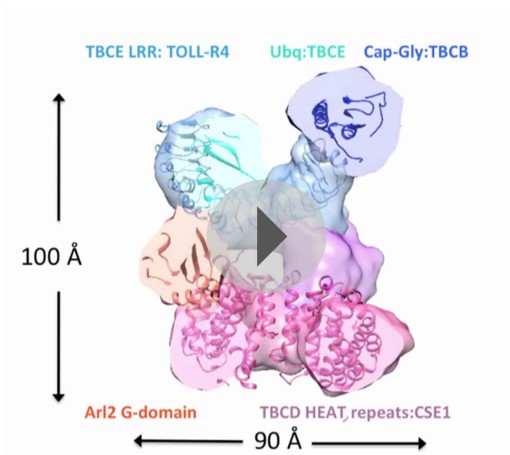

**Video 1.** The video shows a 360° rotation of the raw TBC-DEG-Q37L map (*Figure 4A* ) and a 360° rotation of the segmented TBC-DEG-Q73L map (accompanies *Figure 4*). The color scheme is described in *Figure 3B*. This is followed by a 360° rotation of the raw TBC-DEG-Q37L segmented and coordinate fitted map (*Figure 4D–G*), followed by a clipping view slicing across a segmented and fitted TBC-DEG-Q73L map (*Figure 4G*).

supplement 2B*). We located the TBCE N-terminus near one of the two globular domains at the end of the bow, suggesting that this density is the N-terminal Cap-Gly domain of TBCE (*Figure 4—figure supplement 2B*).

We built a pseudo-atomic model for TBC-DEG using the experimentally verified position for the TBCE Cap-Gly domain, followed by semi-automated docking of Arl2 and structural models for conserved domains of TBCD and TBCE, into the TBC-DEG map (*Figure 4E–G*). We used the structure of Cse1p as a model for TBCD, as they share over 47% sequence identity and contain a similar number of HEAT repeats (*Cook et al., 2005b*). For the TBCE LRR domain, we used the structure of TLR4, which shares 40% sequence similarity with TBCE and contains 14 LLR repeats (*Park et al., 2009b*). We used the Cap-Gly (*Fleming et al., 2013c*) and ubiquitin-like (*Fleming et al., 2013b*) domains of TBCB as models for the equivalent domains of TBCE, and the human Arl2 structure as a model for yeast Arl2 (*Hanzal-Bayer et al., 2002b*). We used two approaches to dock these subunit models into the TBC-DEG map, leading to similar results (see 'Materials and methods'). First, we segmented the TBC-DEG-Q73L map, and docked the atomic models into

segments using the 'fit to segment' feature in UCSF Chimera. Second, we used low-resolution filtered subunit models and successively fit them into an unsegmented map, using the 'fit in map' feature of UCSF Chimera, starting with the largest subunit (TBCD) and ending with the smallest subunit (Arl2) (*Figure 4—figure supplement 3A–C*). The Cse1 model representing TBCD fit well to the floor and thumb segments (UCSF Chimera correlation coefficient 0.71) and its distinct circular-ring and rod shape allowed an unambiguous fit directly into the TBC-DEG map without segmentation. TLR4, representing the central LRR domain of TBCE, fit well to the bow segment of the TBC-DEG map (UCSF Chimera correlation coefficient 0.80) after placement of TBCD (*Figure 4—figure supplement 3B*). The TBCB Cap-Gly structure fit well into the globular end of the bow closest to the TBCE N-GFP density from TBC-DE(N-GFP)G (*Figure 4—figure supplement 2*), while the TBCB ubiquitin-like domain fit well into the other globular end of the bow (UCSF Chimera correlation coefficient 0.82). The overall bow-shaped organization of TBCE LLR with two globular domains at its ends is similar to the TBCE organization in a recent study (*Serna et al., 2015*). Finally, Arl2 fit well to the pillar density located in between the TBCE and TBCD subunits (*Figure 4—figure supplement 3C*). The only region of the TBC-DEG map that was not accounted for in our modeling is the trunk, which likely includes the four to five HEAT repeats in TBCD that are missing from the Cse1 model according to sequence alignment comparison. Thus, our low-resolution structure and model for TBC-DEG support our domain deletion/insertion analysis of TBCD, TBCE, and Arl2 (*Figure 2—figure supplement 1B*), indicating a non-linear assembly of these subunits into a cage-like complex, and demonstrating critical roles for N and C-terminal domains of TBCD, Arl2, and the C-terminus of TBCE in TBC-DEG assembly (*Figure 4H,I*).

## TBC-DEG embraces an αβ-tubulin dimer asymmetrically above its hollow core

To examine the structural basis for TBC-DEG association with αβ-tubulin, we determined a 3D reconstruction for the TBC-DEG-Q73L:αβ-tubulin complex using the approach described above (see 'Materials and methods'). Raw images and reference-free classification show TBC-DEG-Q73L: αβ-tubulin particles adopt a variety of orientations, with a moderate degree of preferred views (*Figure 5—figure supplement 1*). We generated a de novo starting model using common class averages (*Figure 5—figure supplement 1B*, left panel). Using a low-resolution filtered starting

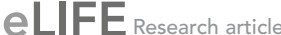

**Figure 5**. TBC-DEG platforms engage the αβ-tubulin dimer asymmetrically, placing it in contact with Arl2 GTPase above the hollow core. (**A**) A refined TBC-DEG-Q73L:αβ-tubulin 3D map shown in three rotated views. The map shows the presence of dual regions at the top of the TBC-DEG cage density. (**B**) A segmented TBC-DEG-Q73L: αβ-tubulin map shown in three rotated views. A dual lobed density (red) assigned to αβ-tubulin is bound by domains at the top side of the TBC-DEG-Q73 cage. Video 3 shows the **A** and **B** views. (**C**) A TBC-DEG-αβ-tubulin linear domain map shown to length scale. TBCD (pink, top panel) is composed of HEAT repeats. TBCE (second panel) includes a Cap-Gly domain (dark blue), a leucine rich repeat (LRR) domain (blue), and a ubiquitin-like domain (cyan). Arl2 (third panel) consists of a G-domain or GTPase fold (orange). αβ-tubulin (red) is shown in the bottom panel. Colors correspond to subunits shown in **D**–**I**. (**D**) A Pseudo-atomic model of the TBC-DEG-Q73L:αβ-tubulin complex showing the interfaces of TBCD, TBCE, and Arl2 engaging the intact αβ-tubulin asymmetrically. The model is built by fitting the densities of TBC-DEG segments as described in *Figure 4* in addition to αβ-tubulin structure into the

*Figure 5. continued on next page*

*Figure 5. Continued*

dual lobed density. (**E**) A 90° vertically rotated view of that shown in **D**. (**F**) A 90° horizontally rotated view of that shown in **D**. *Video 2* shows the **C**–**F** views. (**G**) A central slice view of 90° counterclockwise horizontally rotated view of that shown in **D**. (**H**) Cartoon view of TBC-DEG domain organization comparable to the view shown in **F** (**I**) Cartoon view of TBC-DEG domain organization comparable to the view shown in **G**.

The following figure supplement is available for figure 5:

**Figure supplement 1**. Electron microscopy and 3D reconstruction of the TBC-DEG-Q73L:αβ-tubulin complex.

---

model, we then carried out projection matching and angular reconstitution cycles (see 'Materials and methods'; *Figure 5—figure supplement 1C*, *Table 4*) and then refined a 24 Å TBC-DEG:αβ-tubulin map. The TBC-DEG:αβ-tubulin model projections match the class averages (*Figure 5—figure supplement 1D*). When compared to TBC-DEG-Q73L, the TBC-DEG-Q73L:αβ-tubulin map shows an additional dual-lobed mass on top of the cage, with dimensions of approximately 80 × 40 × 40 Å, which we assigned as αβ-tubulin (*Figure 5A*, *Figure 5—figure supplement 1D*). Although the orientation of the tubulin dimer cannot be determined solely from the electron density map, the TBCE Cap-Gly domain is known to bind the disordered α-tubulin C-terminus (*Akhmanova and Steinmetz, 2008*). This additional information allows us to unambiguously assign the orientation of the bound αβ-tubulin, and fitting of the atomic coordinates of the αβ-tubulin dimer into the dual-lobed density (UCSF Chimera correlation coefficient 0.85) results in a pseudo-atomic model of the full TBC-DEG-Q73L:αβ-tubulin complex (*Figure 5D*).

The pseudo-atomic model of TBC-DEG-Q73L:αβ-tubulin shows the αβ-tubulin dimer embraced by the TBCD thumb and the TBCE LRR domain on its two lateral MT-forming interfaces. In addition, the TBCE Cap-Gly domain is positioned above α-tubulin, and Arl2 contacts β-tubulin from below. The TBCE Cap-Gly domain moves ~20 Å from its position in TBC-DEG-Q73L, shifting closer to the TBCD thumb, while interfacing with the density we assign as α-tubulin. Upon tubulin binding, the TBCD thumb, TBCE LRR, and Arl2 GTPase domains each move ~10 Å closer to the center of the TBC-DEG cage (*Figure 5—figure supplement 1C*). TBC-DEG thus engages three sides of the αβ-tubulin dimer, intimately embracing the individual monomers above its hollow core. The domain connecting the TBCE bow to the Cap-Gly domain becomes disordered upon binding, suggesting that TBCE undergoes a conformational change (*Figure 5D–G*, dark blue). The TBCE ubiquitin domain could not be assigned in this map (*Figure 5D–G*). Importantly, our TBC-DEG:αβ-tubulin model shows that both longitudinal interfaces used in MT protofilament assembly are accessible while tubulin is engaged by TBC-DEG (*Figure 5D–G*). Thus, our structural analysis suggests that the TBC-DEG chaperone organization is likely critical for recognizing α- and β-tubulin monomers during the biogenesis or degradation of αβ-tubulin dimer (*Figure 5H,I*).

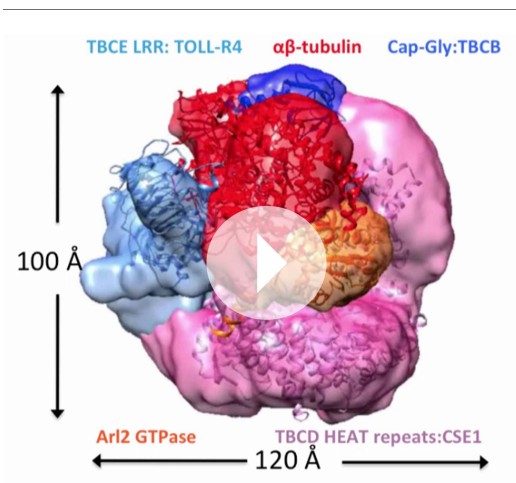

TBCE LRR: TOLL-R4 αβ-tubulin Cap-Gly:TBCB

100 Å

Arl2 GTPase TBCD HEAT repeats:CSE1

120 Å

**Video 2.** The video shows a 360° rotation of the raw TBC-DEG-Q37L:αβ-tubulin map (*Figure 5A*) and a 360° rotation of the segmented TBC-DEG-Q73L:αβ-tubulin map (accompanies *Figure 5*). This is followed by 360° rotation of the overlaid TBC-DEG-Q73L (blue) and TBC-DEG-Q73L:αβ-tubulin maps (red) as shown in *Figure 5—figure supplement 1D*. This is followed by 360° rotation of the raw TBC-DEG-Q37L:αβ-tubulin segmented and coordinate fitted map (*Figure 5D–G*), followed by a clipping view slicing across the segmented and fitted TBC-DEG-Q73L:αβ-tubulin map.

## TBCC C-terminus is a β-helix wedge that catalyzes αβ-tubulin dependent GTP hydrolysis

Sequence alignments suggest that TBCC is a two-domain protein (*Figure 6—figure supplement*

*1E*), with an N-terminal spectrin-like domain (residues 1–99 of 267; *Garcia-Mayoral et al., 2011*), and a C-terminal domain predicted to consist of β-sheets (residues 100–267) that is likely to be responsible for TBCC GAP activity (*Figure 6—figure supplement 1F*; *Kuhnel et al., 2006*; *Mori and Toda, 2013*). To better understand TBCC's interactions with TBC-DEG, we sought to determine TBCC's crystal structure. We crystallized full-length *S. cerevisiae* TBCC and determined a 2.0 Å resolution structure encompassing residues 100–267 (*Figure 6—figure supplement 1A*; see 'Materials and methods'; *Table 5*). Electron density for the TBCC N-terminal domain was absent, indicating it is either disordered or proteolyzed during crystallization. The TBCC C-terminal domain adopts a β-helix fold composed of 13 β-strands arranged in a helical staircase in the shape of a narrow triangular wedge (*Figure 6A–C*). TBCC shows structural homology to retinitis pigmentosa-2 (RP-2) protein (RMSD 1.7 Å; *Figure 6—figure supplement 1C*), a well-studied GAP for the Arl2 paralog Arl3 (*Kuhnel et al., 2006*). In RP2, the β-helix domain binds Arl3 and inserts an 'arginine finger' into the Arl3 active site to stimulate GTP hydrolysis (*Veltel et al., 2008*). TBCC possesses a conserved arginine (Arg186) in the same position (*Figure 6C*, *Figure 6—figure supplement 1D*), which in our structure projects outward from a highly conserved surface (*Figure 6C,D*). In addition, TBCC includes two conserved features: (1) two additional β-strands with an intervening 15-residue loop (residues 220–245) projecting above the β-helix; and (2) a short C-terminal α-helix that folds onto the TBCC β-helix domain (*Figure 5A*). The TBCC loop is rich in conserved hydrophobic and acidic residues, including Phe233, Phe237, Glu240, Glu241, Glu243, and Asp244 (*Figure 6B*). We generated an Arl2:TBCC interface model by superimposing the TBCC and Arl2 structures onto the RP2:Arl3 co-crystal structure (*Figure 5E*; *Veltel et al., 2008*). This model (detailed in *Figure 6—figure supplement 1D*) predicts that TBCC inserts Arg186 into the Arl2 active site to catalyze GTP hydrolysis, while Phe233 and Phe237 in the TBCC loop bind Arl2 hydrophobic residues, and the TBCC acidic residues 240, 241, 243, and 244 project above the Arl2-TBCC interface.

To determine the significance of the unique structural features of TBCC, we measured the effect of their mutation on GTP hydrolysis activity in TBC-DEG. We first removed the TBCC N-terminal spectrin domain to generate TBCC-C (residues 100–267); this mutant showed a 38% decrease in $k_{cat}$ when compared to wild type TBCC (*Table 3*; *Figure 6F*), and lost the robust αβ-tubulin independent activation of GTP hydrolysis (*Figure 3—figure supplement 1F*). This suggests that TBCC's N-terminal domain likely regulates the αβ-tubulin independent affinity of TBCC for TBC-DEG, while the β-helix domain is sufficient for αβ-tubulin dependent GAP activity. Mutation of the TBCC putative arginine finger, Arg186 (R186A), decreased the GTP hydrolysis rate ($k_{cat}$) of TBC-DEG:αβ-tubulin by slightly more than 70% ($k_{cat} = 0.53$ min$^{-1}$ compared to 1.85 min$^{-1}$; *Figure 6G*, *Figure 3—figure supplement 1E*). As removal of the arginine finger is expected to eliminate the GAP activity of TBCC (*Veltel et al., 2008*), the substantial remaining GTP hydrolysis activity with TBCC R186A supports the idea that GTP hydrolysis observed in TBC-DEG:αβ-tubulin:TBCC complexes arises from a combination of Arl2 and αβ-tubulin (*Figure 6G*, *Figure 3—figure supplement 1G*). A TBCC loop-deleted mutant (Δ233-245; residues 233–245 replaced with a six-residue Ser-Gly linker) reduces GTP hydrolysis activity by 82% ($k_{cat} = 0.34$ min$^{-1}$), to a low yet still robust level of activity, similar to the rate of GTP hydrolysis for TBC-DEG:αβ-tubulin without TBCC (*Figure 3—figure supplement 1B*). The TBCC loop deletion is expected to interfere with Arl2 recognition (*Figure 6B*, *Figure 2—figure supplement 1G*). Our structural and biochemical analyses of TBCC suggest its β-helix domain is a non-classical αβ-tubulin dependent GAP that activates Arl2 GTP hydrolysis, and may activate αβ-tubulin to hydrolyze GTP through an unknown mechanism. Residual GTP hydrolysis after specific inactivation of Arl2 GAP activity through the R186A and Δ233-245 TBCC mutants suggests that a secondary GTPase remains robustly active in the TBC-DEG:αβ-tubulin:TBCC complex. This GTPase is likely αβ-tubulin itself; however, which GTPase site (N or E-site) is becoming activated, and which mechanism is behind its activation, both remain to be determined.

## TBCC β-helix wedge interfaces with Arl2 and αβ-tubulin dimer in the TBC-DEG chaperone

To gain insight into the TBCC GAP mechanism, we determined 3D reconstructions for TBC-DEG-Q73L:αβ-tubulin:TBCC ternary complexes using negative-stain EM and single particle image analysis (see 'Materials and methods'). We used the TBC-DEG-Q73L complex to guarantee 100% stoichiometric TBCC binding in the TBC-DEG-Q73L:αβ-tubulin:TBCC ternary complex to ensure its

**Table 5**. Crystallographic statistics table for TBCC structure determination

| | TBCC native | TBCC Pt-peak | TBCC Pt-inflection | TBCC Pt-remote |
|---|---|---|---|---|
| **Data collection** | | | | |
| Resolution range (Å) | 34.90–2.00 (2.07–2.00)* | 41.80–2.18 (2.30–2.18)* | 35.01–2.18 (2.30–2.18)* | 31.94–2.18 (2.30–2.18)* |
| Space group | $P\,4_3$ | $P\,4_3$ | $P\,4_3$ | $P\,4_3$ |
| Wavelength (Å) | 0.9795 | 1.0715 | 1.0719 | 0.9537 |
| Unit cell (Å): $a$, $b$, $c$ | 69.79, 69.79, 78.18 | 70.03, 70.03, 77.95 | 70.03, 70.03, 77.95 | 70.03, 70.03, 77.95 |
| Total reflections | 193,620 | 198,772 | 198,980 | 199,576 |
| Unique reflections | 25,377 (2504)* | 19,716 (2871)* | 19,752 (2879)* | 19,748 (2877)* |
| Average mosaicity | 0.29 | 0.40 | 0.40 | 0.42 |
| Anomalous multiplicity | – | 5.1 (5.0)* | 5.1 (5.0)* | 5.2 (5.1)* |
| Multiplicity | 7.6 (7.6)* | 10.1 (10.0)* | 10.1 (9.9)* | 10.1 (10.1)* |
| Anomalous completeness (%) | – | 100.0 (100.0)* | 100.0 (100.0)* | 100.0 (100.0)* |
| Completeness (%) | 100.0 (100.0)* | 100.0 (100.0)* | 100.0 (100.0)* | 100.0 (100.0)* |
| $<I/\sigma\,(I)>$ | 13.4 (2.7)* | 28.6 (9.0)* | 28.4 (8.8)* | 28.1 (8.5)* |
| $R_{merge}$† | 0.082 (0.72)* | 0.046 (0.23)* | 0.046 (0.23)* | 0.046 (0.25)* |
| f′ | – | 17.40 | 23.23 | 4.70 |
| f″ | – | 15.67 | 10.29 | 11.30 |
| **Structure refinement** | | | | |
| $R_{work}$ | 0.20 (0.26)* | – | – | – |
| $R_{free}$ | 0.24 (0.28)* | – | – | – |
| Molecules per asymmetric unit | 2 | – | – | – |
| Number of atoms | 2744 | – | – | – |
| Protein residues | 329 | – | – | – |
| Number of water molecules | 93 | – | – | – |
| RMS bond lengths (Å) | 0.007 | – | – | – |
| RMS bond angles (°) | 1.00 | – | – | – |
| Ramachandran favored (%) | 96.0 | – | – | – |
| Ramachandran allowed (%) | 0.0 | – | – | – |
| Ramachandran outliers (%) | 0.0 | – | – | – |
| Clashscore | 4.6 | – | – | – |
| **Mean $B$ values (Å²)** | | | | |
| Overall | 50.4 | – | – | – |
| Main-chain atoms | 46.2 | – | – | – |
| Side-chain atoms | 54.6 | – | – | – |
| Solvent | 49.4 | – | – | – |

*Numbers represent the highest-resolution shell.

†$R_{merge} = \Sigma_{hkl}\Sigma_i|I_i(hkl) - I_{av}(hkl)|/\Sigma_{hkl}\Sigma_i I_i(hkl)$.

full occupancy in the structural studies. Raw images and reference-free classification indicate that the hollow core of the cage becomes largely occupied in the ternary complex. This conformation is distinct in appearance from the previous conformations (*Figure 7—figure supplement 1B*). We used angular reconstitution and refinement to generate a 24 Å TBC-DEG-Q73L:αβ-tubulin:TBCC map (*Figure 7A*, *Table 4*), and the model projections match well to the reference-free class averages (*Figure 7—figure supplement 1E*). The ternary complex map was then interpreted with respect to the TBC-DEG-Q73L and TBC-DEG-Q73L:αβ-tubulin maps (*Figure 7B*). The ternary complex map shows an additional wedge-shaped density inside the hollow core of the TBC-DEG cage, which we

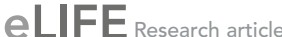

**Figure 6**. TBCC catalytic C-terminal domain x-ray structure suggests a TBCC-Arl2 binding interface to dissect the Arl2 contribution TBC-DEG GTP hydrolysis. (**A**) The 2.0 Å x-ray structure of the TBCC C-terminal β-helix domain (100–267) in two rotated views. β-sheets (red) form a narrow helical structure in which turns (green) lie at the ends and a large conserved and structured loop (purple) is presented on top of the structure. (**B**) A close-up view of the TBCC conserved residues in the structured loop showing the hydrophobic (Phe233, 237) and acidic (Glu240, 241, 243 and Asp244) residues. (**C**) A close-up view of the Arl2 catalytic interface showing Glu184, Arg186, and Phe164. (**D**) TBCC β-helix surface conservation showing the high conservation of the Arl2 catalytic site on one side of the TBCC β-helix domain. The left panel, front view, and 90° rotated view are shown. A color key gradient describing the conservation is shown below, with purple denoting highest and cyan denoting lowest conservation. (**E**) An interface model for

*Figure 6. continued on next page*

*Figure 6. Continued*

TBCC-β-helix domain-Arl2, based on a superimposition onto the RP2-Arl2 structures, which is described in detail in *Figure 6—figure supplement 1*, panel **D**. The model shows Arg186 to be the arginine finger activating GTP hydrolysis in Arl2. The Arl2 Gln73 interacts with a water molecule required for GTP nucleotide (shown in blue) during hydrolysis. The model shows the TBCC loop resides above Arl2 during the catalytic interface. (**F**) Steady-state GTP hydrolysis activity of TBC-DEG+αβ-tubulin+TBCC β-helix C-terminal domain 100–267, TBCC-C (DEG+TBCC-C +αβ-tub, brown) compared to TBC-DEG+αβ+tubulin+wild type TBCC (DEG+wtTBCC+αβ-tub, blue), showing the TBCC C-terminal domain is sufficient for GTP hydrolysis. TBC-DEG alone (DEG, shown in red) has very low basal GTP hydrolysis activity. Parameters are described in *Table 3*. (**G**) Steady-state GTP hydrolysis activity of TBC-DEG+ αβ-tubulin+TBCC Arg186Ala (DEG+R186+αβ-tub, orange) compares well to wild type TBCC+TBC-DEG (DEG+ wtTBCC; green), suggesting similar GTP hydrolysis parameters. TBC-DEG+αβ+tubulin+TBCC (DEG+Δ233-245+ αβ-tub, pink) shows a similar defect in GTP hydrolysis. TBC-DEG+TBCC+αβ-tubulin (DEG+wtTBC+αβ-tub; blue) has a twofold higher GTP hydrolysis rate. Parameters are described in *Table 3*.

The following figure supplement is available for figure 6:

**Figure supplement 1**. TBCC C-terminal domain is a β-helix structure with dual interfaces.

assign to the TBCC β-helix domain. A second density is observed engaging the αβ-tubulin at its intra-dimer interface, located in proximity to the TBCE Cap-Gly domain densities in previous maps (*Figure 7A,B*). The TBCC C-terminal domain-Arl2 GTPase interface lies directly below the intra-dimer interface of tubulin. We generated a pseudo-atomic model for TBC-DEG-Q73L:αβ-tubulin:TBCC by fitting atomic coordinates for TBCC-N spectrin in the additional density along the TBC-DEG floor, and fit the C-terminal β-helix domain to the wedge density (*Figure 7C*). The TBCC β-helix docked well to the wedge density engaging the Arl2 GTPase (UCSF Chimera correlation coefficient 0.85), and this fit

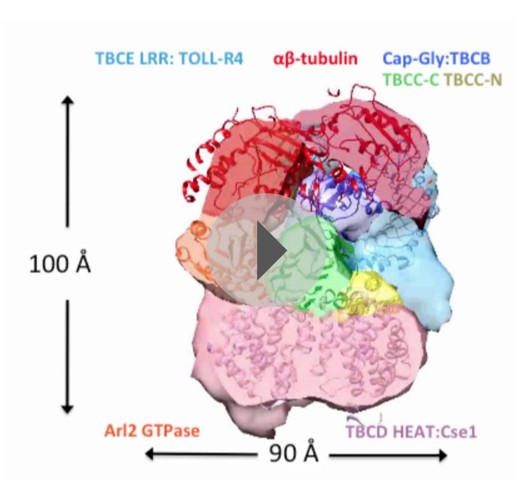

**Video 3.** The video shows a 360° rotation of the raw TBC-DEG-Q37L:αβ-tubulin:TBCC map (*Figure 7A*) followed by a 360° rotation of the overlaid TBC-DEG-Q73L:αβ-tubulin:TBCC map (blue) with the TBC-DEG-Q73L:αβ-tubulin map (red) as shown in *Figure 7—figure supplement 1D* (accompanies *Figure 7*). This is next followed by a 360° rotation of the segmented TBC-DEG-Q73L:αβ-tubulin:TBCC map (accompanies in *Figure 7*), followed by a 360° rotation of the TBC-DEG-Q37L: αβ-tubulin:TBCC segmented and fitted map (*Figure 7D–G*), and followed by a clipping view slicing across the segmented and fitted TBC-DEG-Q73L: αβ-tubulin:TBCC map.

matches the conformation of our homology model for the TBCC-Arl2 binary complex (*Figure 6E*). Strikingly, despite the low resolution of our maps, we find that the αβ-tubulin dimer adopts a unique conformation in the ternary complex. The intact αβ-tubulin dimer did not fit into this density in the ternary complex map. Therefore, α and β-tubulin models were fit individually (UCSF Chimera correlation coefficients 0.55 and 0.67 for α and β-tubulin, respectively; *Figure 7H*). The αβ-tubulin intra-dimer interface is wedged open by a 20 Å-wide globular density, which we assigned to be the TBCE Cap-Gly domain due to its physical proximity in the TBC-DEG and TBC-DEG: αβ-tubulin maps. We suggest that the TBCE Cap-Gly domain is repositioned by 10 Å in the ternary complex, wedging between α- and β-tubulin. TBCC C-terminal β-helix and its loop lie directly below the intra-dimer interface. At the current resolution it remains unclear how the αβ-tubulin dimer is modified and we require higher resolution studies to understand its conformation and the positioning of TBC-DEG domains in the ternary complex. Our structural analysis supports the biochemical finding that TBCC is an αβ-tubulin dependent non-classical GAP for Arl2. Our ternary complex map suggests that TBCC Arg186 activates the Arl2 GTPase while engaging αβ-tubulin at its intra-dimer interface via the extended loop. Arl2 GTP hydrolysis leads to

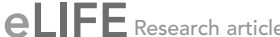

**Figure 7.** A TBC-DEG:αβ-tubulin:TBCC ternary complex structure shows TBCC engages both Arl2 and αβ-tubulin dimer, deforming its intra-dimer interface. (**A**) A refined 24 Å TBC-DEG-Q73L:αβ-tubulin:TBCC 3D map shown in three rotated views. The map shows conformational changes in tubulin density and the presence of new densities in the hollow core of the cage. (**B**) A segmented 24 Å TBC-DEG-Q73L:αβ-tubulin:TBCC map shown in three rotated views. The tubulin dimer density (red) is deformed by two new densities: a TBCC wedge shaped density engages the Arl2 interface (green), and a second density (cyan) is wedging between the two αβ-tubulin dimer lobes (red). (**C**) A TBC-DEG:αβ-tubulin:TBCC linear domain map shown to length scale. TBCD (pink, top panel) is composed of HEAT repeats. TBCE (second panel) includes a Cap-Gly domain (dark blue), a leucine rich repeat (LRR) domain (blue), and a ubiquitin-like domain (cyan). TBCC consists of a spectrin domain (yellow) and a C-terminal β-helix domain (green) (described in **Figure 5**), and Arl2 (third panel) consists of a GTPase fold (orange). αβ-tubulin (red) is shown in the bottom panel. Colors correspond to subunits shown in **D–I**. (**D**) A pseudo-atomic model of the TBC-DEG-Q73L:

*Figure 7. continued on next page*

*Figure 7. Continued*

αβ-tubulin:TBCC ternary complex showing the interfaces of the TBCC β-helix catalytic domain (described in *Figure 5*, green) engaging Arl2 (orange) on top of TBCD (pink) while bound by the TBCE LRR bow (blue), while the α and β-tubulins are wedged by the ubiquitin domain (cyan). The α and β-tubulin coordinates were fit individually due to deformation in the tubulin intra-dimer interface in this map. The TBCC N-terminal spectrin domain (yellow) was fit into a density added to the floor segment. (**E**) A 90° vertically rotated view of that shown in **D**. (**F**) A 90° horizontally rotated view of that shown in **D**. (**G**) A central slice view of a 90° counterclockwise horizontally rotated view of that shown in **D**. The TBCC C-terminal catalytic domain engages Arl2 and binds the αβ-tubulin at the deformed intra-dimer interface with its unique loop (pink ribbon). *Video 3* shows the **C**–**F** views. (**H**) Comparison of αβ-tubulin conformation based on αβ-tubulin coordinates fit into the αβ-tubulin density from the TBC-DEG-Q73L:αβ-tubulin map shown in the left panel (−TBCC) compared to the αβ-tubulin coordinates fit into the αβ-tubulin density in the TBC-DEG-Q73L:αβ-tubulin:TBCC map shown on the right (+TBCC), showing the conformational change at its intra-dimer interface that is associated with TBCC binding. (**I**) Cartoon view of TBC-DEG domain organization comparable to the view shown in **E**. (**J**) Cartoon view of TBC-DEG domain organization comparable to the view shown in **F**.

The following figure supplement is available for figure 7:

**Figure supplement 1**. Electron microscopy and 3D reconstruction of the TBC-DEG-Q73L:αβ-tubulin:TBCC complex.

---

a conformational change that involves a well-documented rotation of its conserved N-terminal helix (*Figure 6—figure supplement 1E*; *Veltel et al., 2008*); this conformational change may reposition the associated TBCE Cap-Gly domain to deform αβ-tubulin or activate αβ-tubulin GTP hydrolysis at its N-site (*Figure 6—figure supplement 1E*).

## Expression of a GTP-locked Arl2 mutant induces severe defects in MT dynamics in vivo

To determine the roles of the TBC-DEG chaperone in regulating MT dynamics and function, we introduced the Q73L mutation into the Arl2 ortholog in budding yeast, Cin4 (*cin4-Q73L*), and observed its effects on MT function and dynamics. First, we tested whether *cin4-Q73L* sensitizes cells to the MT depolymerizing drug benomyl. Wild type and Arl2-deleted (*cin4Δ*) yeast cells were transformed with plasmids containing the *cin4-Q73L* mutant, wild type *CIN4*, or no protein (empty vector) under a galactose-inducible promoter. Consistent with previous studies, *cin4Δ* null mutants exhibit hypersensitivity to benomyl that is rescued by expression of wild type *CIN4* (*Stearns, 1990*; *Figure 8A*). In contrast, *cin4-Q73L* expression elicits dominant hypersensitivity to benomyl; ectopic expression of *cin4-Q73L* sensitized both wild type and *cin4Δ* cells to benomyl (*Figure 8B*). Our genetic rescue experiments suggest that dominant MT function defects are induced when Arl2/*cin4-Q73L* is overexpressed in native or Arl2-deleted cells.

Next, we examined the dynamics of GFP-labeled MTs in yeast cells. We compared the effects of transient *cin4-Q73L* expression in wild type cells to mutant cells with *cin4-Q73L* constitutively expressed from the chromosomal locus (*Figure 8C*). Arl2/*cin4-Q73L* expression decreased the number of MTs per cell, and this effect scaled with duration of *cin4-Q73L* expression (*Figure 8H*; *Video 4*, *Video 7*). A 90 min pulse of ectopic *cin4-Q73L* expression decreased the number of astral MTs (aMTs) in wild type cells. This effect was exacerbated after 180 min of expression (*Figure 8C,H*). Mutant cells constitutively expressing *cin4-Q73L* exhibited the strongest loss of MTs (*Figure 8C,H*). Analysis of individual aMT lengths in time-lapse imaging revealed striking changes in MT dynamics. After a 90 min pulse of ectopic *cin4-Q73L*, aMTs were longer and exhibited slower disassembly, decreased rescue frequency, and increased pauses compared to those observed in wild type yeast (*Figure 8D–G*; *Table 6*; *Videos 5, 6*). Cells expressing constitutive *cin4-Q73L* also exhibited decreased rescue frequency and increased pauses. However, aMTs were slightly but significantly shorter than those in wild type cells; in contrast, MT disassembly rates were not significantly different. In conclusion, our studies suggest that the previously well-characterized phenotypes of TBC protein inactivation (*Hoyt et al., 1990*; *Radcliffe et al., 1999*; *Antoshechkin and Han, 2002*; *Jin et al., 2009*) may be a result of soluble αβ-tubulin regulation defects leading to aberrant MT dynamics.

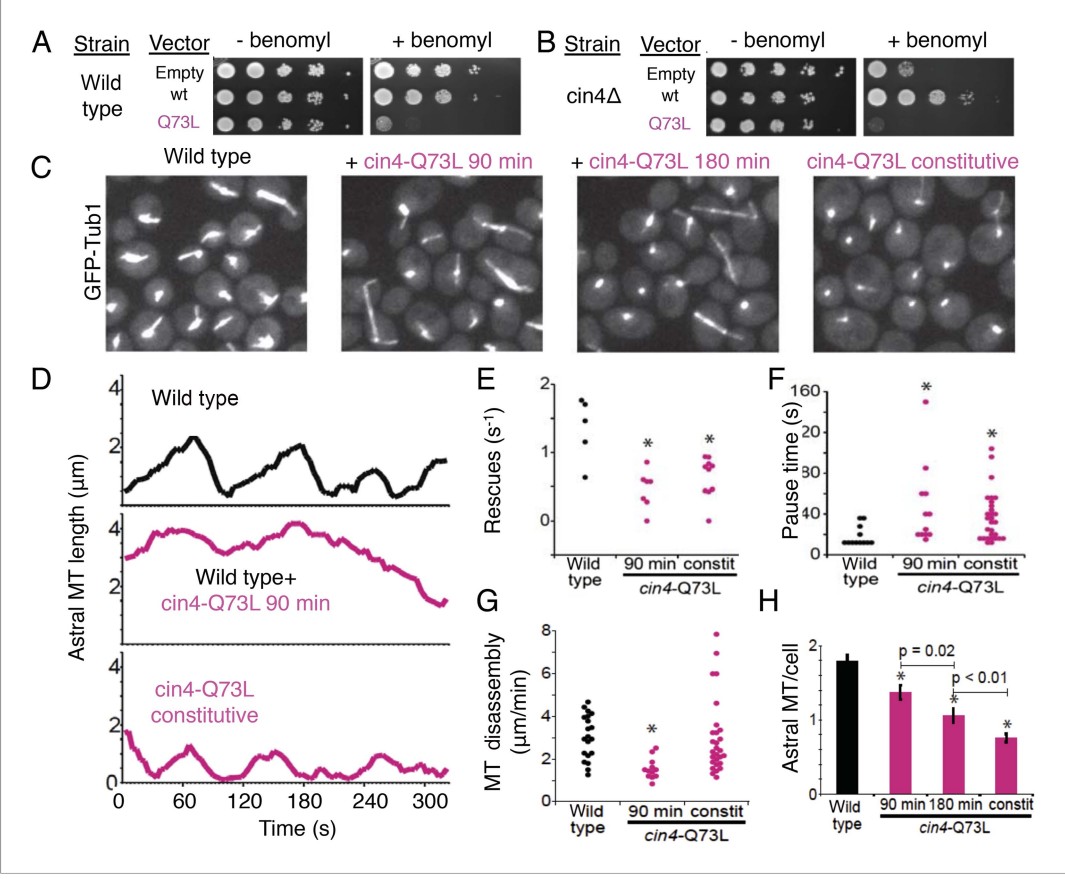

**Figure 8**. Introducing the Arl2 GTP-locked Q73L mutation induced pausing of dynamic MTs in vivo. (**A**) Expression of *cin4*-Q73L elicits dominant benomyl sensitivity and MT polymerization defects. Wild type or *cin4*Δ mutant cells transformed with plasmids expressing the indicated genes under galactose-inducible promoters were plated on inducing media without benomyl or with 10 µg/ml benomyl, and grown at 30°C for 4 days. (**B**) Expression of *cin4*-Q73L interferes with MT dynamics and activates pausing. Right panel, wild type cells expressing MT labeled with Tub1-GFP. Strain: yJM0562. Second panel, wild type yeast cells expressing Tub1-GFP transformed with a *cin4*-Q73L expression plasmid and treated with galactose to induce expression for 90 min before imaging. Third panel, a separate population of wild type cells transformed with a *cin4*-Q73L expression plasmid and induced for 180 min before imaging. Right panel, cells constitutively expressing *cin4*-Q73L from the chromosomal locus. Images are 2D projections of 13 Z planes separated by 400 nm. (**C**) Representative raw fields of yeast cells in each condition with MTs labeled with GFP-tub. *Videos 4–7* accompany these panels. (**D**) Representative lifeplots of astral MT dynamics in wild type cells (top panel), wild type cells expressing *cin4*-Q73L for 90 min (middle panel), and *cin4*-Q73L mutants (bottom panel). Astral MT length was measured over time by plotting the distance between the spindle and the distal end of the MT. Strains: wild type, yJM0562; *cin4*-Q73L, yJM1375. (**E**) MT rescue frequencies from time-lapse imaging of wild type (n = 5), wild type expressing *cin4*-Q73L for 90 min (n = 7), and constitutive *cin4*-Q73L mutant (n = 10) cells. Asterisks indicate statistical significance (p<0.01) determined by t-test, compared to wild type. Strains: wild type, yJM0562; *cin4*-Q73L, yJM1375. (**F**) Durations of pause events from time-lapse imaging of wild type (black), wild type expressing *cin4*-Q73L for 90 min, and constitutive *cin4*-Q73L mutant cells (shown in pink). Asterisks indicate statistical significance (p<0.01) determined by t-test, compared to wild type. Strains: wild type, yJM0562; *cin4*-Q73L, yJM1375. (**G**) Average MT disassembly rates from time-lapse imaging of wild type (black), wild type expressing *cin4*-Q73L for 90 min, and constitutive *cin4*-Q73L mutant cells (shown in pink). Asterisks indicate statistical significance (p<0.01) determined by t-test, compared to wild type. Strains: wild type, yJM0562; *cin4*-Q73L, yJM1375. (**H**) Average length of astral MTs (aMT) per yeast cell measured, for wild type (black), wild type expressing *cin4*-Q73L for 90 min, and constitutive *cin4*-Q73L mutant cells (shown in pink). Asterisks indicate statistical significance (p<0.01) determined by t-test, compared to wild type. Strains: wild type, yJM0562; *cin4*-Q73L, yJM1375.

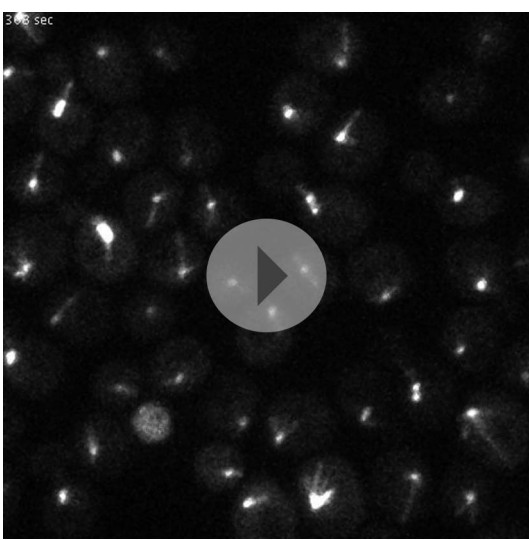

**Video 4.** Microtubule dynamics in wild type cells (accompanies *Figure 8C*). Time-lapse images of wild type cells expressing Tub1-GFP were captured on a spinning disk confocal microscope at 4 s intervals for 10 min. Each image represents a composite of 13 planes separated by 400 nm. Video plays at 60 times real time. Strain yJM0562.

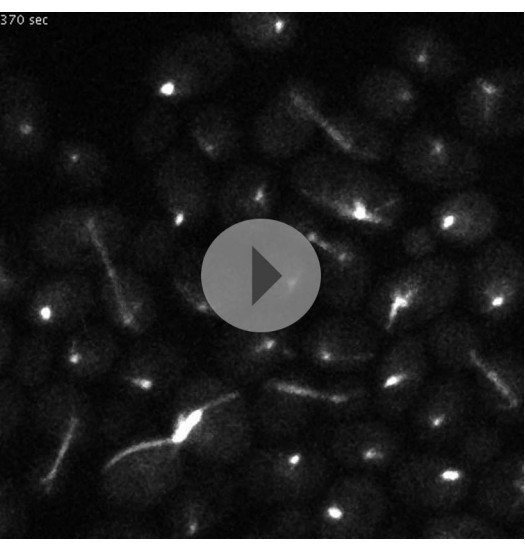

**Video 5.** Microtubule dynamics after 90 min of *cin4*-Q73L expression (accompanies *Figure 8C*). Time-lapse images of wild type cells expressing Tub1-GFP containing a *cin4*-Q73L expression plasmid after 90 min of induction. Images were captured on a spinning disk confocal microscope at 5 s intervals for 10 min. Each image represents a composite of 15 planes separated by 400 nm. Video plays at 60 times real time. Strain yJM0562, with plasmid pJM0231.

## Discussion

Here we describe the molecular mechanisms and enzymatic activity of the tubulin cofactors and the Arl2 GTPase, and reveal their shared role as a critical multi-subunit chaperone that regulates the soluble pool of αβ-tubulin in the cytoplasm (*Figure 9B*). We show that TBCD, TBCE, and the Arl2 GTPase form a chaperone core (TBC-DEG), which sequentially binds intact soluble αβ-tubulin and TBCC to activate the Arl2 GTPase and possibly influence the αβ-tubulin conformation and GTPase activity. In addition to functioning in αβ-tubulin dimer biogenesis, we suggest that this chaperone system may also regulate the activity of the existing pool of soluble αβ-tubulin dimers in the cytoplasm.

Our biochemical and structural analyses suggests that TBC-DEG grasps individual αβ-tubulin dimers via TBCD and TBCE, and then catalytically manipulates their conformation through the GTPase activity of Arl2 (*Figure 9*). This manipulation likely drives both biogenesis and degradation of the αβ-tubulin dimer. We find that TBCC acts as a true GAP for Arl2, likely inserting the arginine finger residue Arg186 into the Arl2 active site to stimulate GTP hydrolysis. We find that full GAP activity depends on the binding of αβ-tubulin onto the TBC-DEG chaperone, suggesting that TBCC's affinity for TBC-DEG is increased upon αβ-tubulin binding to the chaperone. We suggest that this affinity enhancement may be mediated by the extended acidic loop of TBCC, which is positioned close to the bound αβ-tubulin dimer in our pseudo-atomic model of the ternary complex, and mutation of which eliminates tubulin-dependent GAP enhancement. Finally, we observe modest but reproducible GTP hydrolysis both in the absence of TBCC and in an arginine finger-defective TBCC R186A mutant, suggesting that in addition to Arl2, a tubulin GTPase site (either N or E-site) may become active when bound to the TBC-DEG chaperone.

An important question is how TBC-DEG, through Arl2 GTPase activity, mediates both assembly and activation of the αβ-tubulin dimer, and potentially also its dissolution into α- and β-tubulin monomers. When TBC-DEG grasps α- and β-tubulin, the tubulin dimer interface is positioned directly above the Arl2 and its associated TBCC GAP. We suggest these binding interfaces are critical for setting the αβ-tubulin configuration during biogenesis (*Figure 9A*). The TBC-DEG catalytic activity is likely required to overcome the high affinity of α- for β-tubulin within tubulin dimers. Previous studies have demonstrated that αβ-tubulin dimer dissociation is an extremely slow and unfavorable biochemical

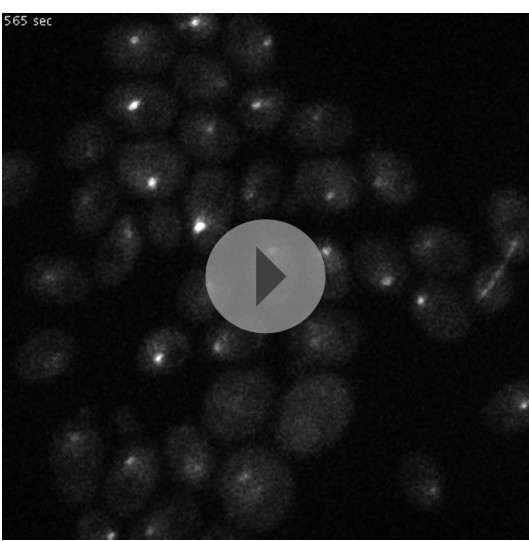

**Video 6.** Microtubule dynamics after 180 min of *cin4*-Q73L expression (accompanies *Figure 8C*). Time-lapse images of wild type cells expressing Tub1-GFP containing a *cin4*-Q73L expression plasmid after 180 min of induction. Images were captured on a spinning disk confocal microscope at 5 s intervals for 10 min. Each image represents a composite of 15 planes separated by 400 nm. Video plays at 60 times real time. Strain yJM0562, with plasmid pJM0231.

EM maps do not yet show how the Arl2 GTPase chaperone enzymatic activity, and what domains in

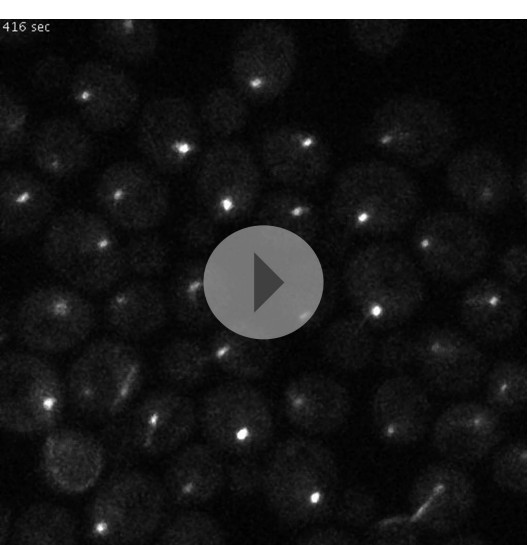

**Video 7.** Microtubule dynamics in *cin4*-Q73L mutant cells (accompanies *Figure 8C*). Time-lapse images of cells constitutively expressing *cin4*-Q73L from the chromosomal locus and Tub1-GFP were captured on a spinning disk confocal microscope at 4 s intervals for 10 min. Each image represents a composite of 13 planes separated by 400 nm. Video plays at 60 times real time. Strain yJM1375.

process that requires an external energy source (*Caplow and Fee, 2002*). The TBC-DEG chaperone system likely provides the required energy through TBCC-mediated Arl2 GTP hydrolysis to power the dissociation of αβ-tubulin dimers. The TBCC αβ-tubulin binding interface ensures that catalysis is activated more efficiently upon αβ-tubulin binding to TBC-DEG.

We suggest a revised 'cycling chaperone' paradigm for tubulin cofactors and Arl2: (1) TBC-DEG forms a nascent αβ-tubulin dimer through TBCA and TBCB tubulin monomer delivery, or may bind an already formed soluble αβ-tubulin dimer; and (2) TBCC binds the TBC-DEG-αβ-tubulin complex and activates the Arl2 GTPase in an αβ-tubulin dependent manner, driving TBCD and TBCE conformational changes to manipulate αβ-tubulin, leading either to its degeneration or release (*Figure 9B*). The TBCC αβ-tubulin dependent GAP may ensure the correct αβ-tubulin configuration and activate Arl2 GTP hydrolysis. The Arl2 GTPases were shown to undergo dramatic conformational change upon GTP hydrolysis (*Figure 6—figure supplement 1E*), which likely drives tubulin cofactor chaperone catalysis. However, at the current resolution, our current conformational change catalyzes the TBC-DEG chaperone activity, and what domains in TBC-DEG mediate the dissociation of the αβ-tubulin dimer into monomers.

In yeast, expression of tubulin cofactor chaperones with Arl2 locked in a GTP-like state (*cin4*-Q73L) leads to severe defects in MT dynamics, characterized by increase in pauses and loss of MT rescues. This regulation may involve soluble αβ-tubulin recycling, biogenesis, and degradation; this activity requires TBCC GAP activity, and an active Arl2 GTPase, where their mutations lead to defects in MT function (*Bhamidipati et al., 2000*; *Fleming et al., 2000*; *Radcliffe et al., 2000*). We postulate that the tubulin cofactor/Arl2 chaperone activity may cumulatively counteract a slow decay in the soluble αβ-tubulin pool (*Figure 9*). Purified soluble tubulin has a well-documented rapid decay if not polymerized into MTs, although the biochemical nature of this decay remains unknown (*Prasad et al., 1986*; *Sackett et al., 1990*; *Sackett and Lippoldt, 1991*; *Elie-Caille et al., 2007*). TBC-DEG chaperones are critical for MT cytoskeletal dynamics in vivo as they may actively remove decaying αβ-tubulin by degradation. GTP hydrolysis within the tubulin dimer, coupled to MT dynamics, may lead to decay in αβ-tubulin, and thus TBC-DEG chaperones may be required to revitalize the αβ-tubulin dimer pool to support

**Table 6**. Microtubule (MT) dynamics in *cin4*-Q73L mutant yeast cells

| | MT length (µm) | Assembly rate (µm/min) | Assembly duration (s) | Disassembly rate (µm/min) | Disassembly duration (s) | Catastrophes (min⁻¹) | Rescues (min⁻¹) | Pause duration (s) |
|---|---|---|---|---|---|---|---|---|
| Wild type | 0.90 ± 0.02 | 1.4 ± 0.06 | 46 ± 3 | 3.06 ± 0.24 | 27 ± 2 | 0.90 ± 0.12 | 1.5 ± 0.18 | 18 ± 3 |
| Wild type + *cin4*-Q73L (90 min) | **2.69 ± 0.05** | 1.4 ± 0.18 | 58 ± 12 | **1.56 ± 0.12** | **53 ± 10** | 0.66 ± 0.12 | **0.48 ± 0.12** | **36 ± 11** |
| *cin4*-Q73L | **0.71 ± 0.02** | 1.5 ± 0.06 | 41 ± 4 | 3.00 ± 0.36 | 24 ± 4 | 0.78 ± 0.06 | **0.66 ± 0.12** | **37 ± 4** |

Values are mean ± SEM of measurements from at least five cells imaged for 600 s. Values in boldface are significantly different from wild type (p<0.05), determined by t-test.

MT dynamics. Locking Arl2 in a GTP state via the Q73L mutation likely inhibits this regulation, leading to a loss in MT polymerization capacity as has been observed in vivo in many systems (*Bhamidipati et al., 2000*; *Tian et al., 2010*; *Mori and Toda, 2013*).

Inactivating Arl2 GTPase leads to inhibition of TBC-DEG catalytic activity and inhibition of TBC-DEG:αβ-tubulin:TBCC ternary complex disassembly (*Figure 3A,B*), and in vivo, shows a rapid decrease in MT polymerization rate and increase in MT pausing. Our in vivo analysis provides compelling evidence for the critical role of TBC-DEG in MT function. We suggest that this chaperone activity is critical for MT dynamics through an effect on the health of the soluble pool of αβ-tubulin. One possibility for the toxic effect of the Arl2 GTP-locked mutation may be the inability of soluble tubulin to cycle through the TBC-DEG chaperone, leading to either sequestration of αβ-tubulin dimer on the complex, or the release of partially dissociated soluble αβ-tubulin dimers, which then induce toxic effects on MT dynamics. In cells, TBC-DEG chaperones are at much a lower concentration of soluble αβ-tubulin, suggesting that TBC-DEG exerts its effects on a relatively small population of αβ-tubulin in the soluble cytoplasmic pool; this idea is supported by previous studies (*Bhamidipati et al., 2000*; *Mori and Toda, 2013*). Increasing concentration of these regulators may catalyze extensive soluble αβ-tubulin degradation, thereby interfering with MT polymerization (*Figure 9B*). The regulation of the soluble αβ-tubulin pool by removing poorly active αβ-tubulin from the pool or adding new fresh αβ-tubulin to the system, may result in more polymerization-competent soluble αβ-tubulin and may enhance MT dynamics. Locking Arl2 in the GTP state (Q73L) leads to profound changes in MT dynamics leading to a pausing phenotype in vivo, and likely leads to the mitotic defects previously observed (*Fleming et al., 2000*; *Radcliffe et al., 2000*).

Many facets of this paradigm remain to be studied in the future: for example, how is αβ-tubulin released from TBC-DEG once assembled or repaired? How do TBCA-β-tubulin and TBCB-α-tubulin complexes interface with each other and/or TBC-DEG to initially form the αβ-tubulin dimer? How are decaying αβ-tubulin dimers recognized by TBC-DEG for recycling and degradation? How would this chaperone system drive αβ-tubulin biogenesis in the presence of a concentrated soluble αβ-tubulin pool in the cytoplasm?

## Conclusions

We provide a revised paradigm for the assembly, biochemical activity, and organization of the well-conserved tubulin cofactors and Arl2 GTPase as a cage-like chaperone that catalytically alters tubulin dimers in the cytoplasm, powered by GTP hydrolysis. The GTPase activity of Arl2 is central to power and gate these chaperones, while tubulin cofactors TBCD and TBCE mediate molecular recognition of α- and β-tubulin in the heterodimer (*Lewis et al., 1997*; *Tian and Cowan, 2013*). The concept that tubulin cofactors and Arl2 function together as a catalytic chaperone is consistent with long-standing genetics and cell biology studies indicating that their concentration is critical for proper MT dynamics and MT homeostasis. These chaperones represent a new MT regulatory pathway that may enhance MT dynamics by improving the activities of individual soluble αβ-tubulin dimers in the cytoplasmic pool. This regulation is likely critical for the homeostasis of the MT cytoskeleton in eukaryotes, which is underscored by human disorders related to tubulin cofactor mutations.

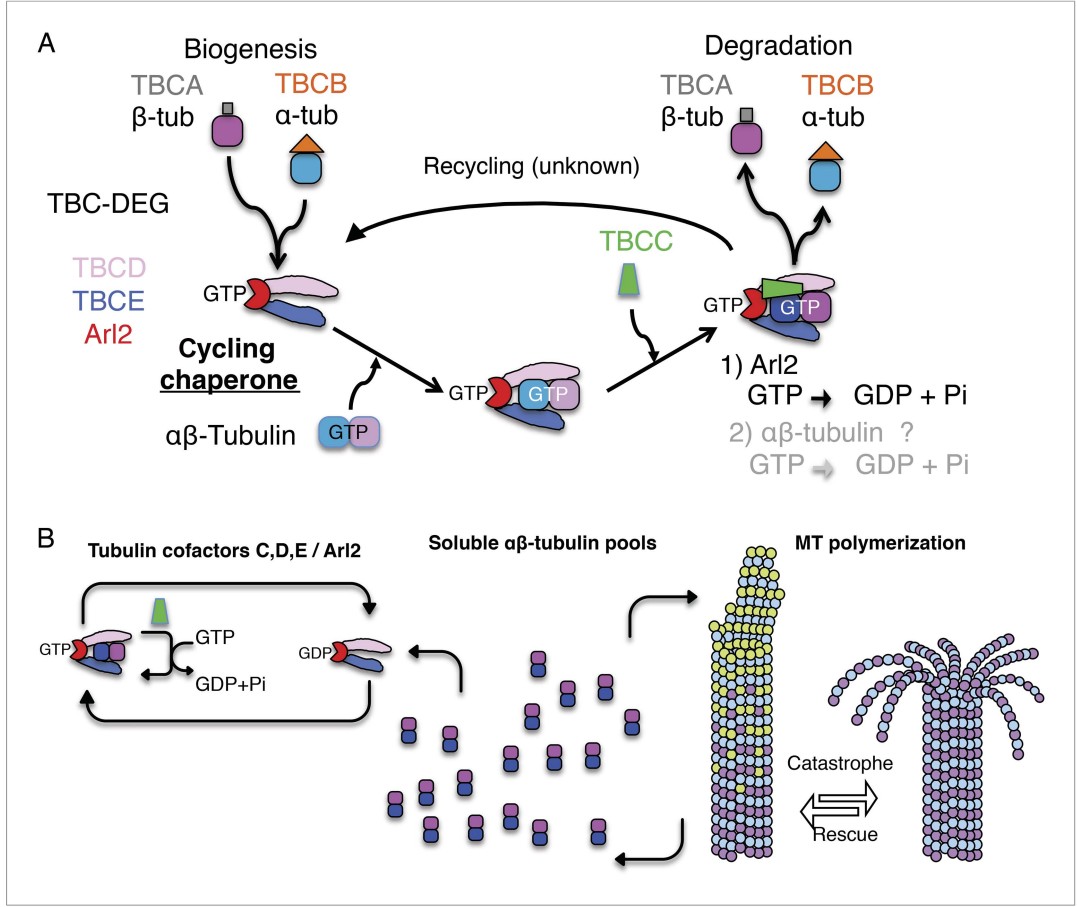

**Figure 9**. A revised scheme for tubulin factors and Arl2 as a chaperone multi-subunit machine in regulating soluble αβ-tubulin. (**A**) Revised paradigm, based on data from this study for tubulin cofactors and Arl2 as a multi-subunit chaperone that cycles to regulate soluble αβ-tubulin through GTP hydrolysis catalytic cycles, while providing sites for α and β-tubulin dissociation for biogenesis and degradation. Through this model, dual GTP hydrolyses in Arl2 and possibly in αβ-tubulin sequentially induce tubulin dimer dissociation without release. We suggest that αβ-tubulin is reassembled and then released. (**B**) An overall model for how TBC-DEG/TBCC activity cycles may regulate slowly decaying tubulin by binding along the TBC-DEG platform, recruiting TBCC, and then dissociating α and β-tubulin from each other, without dissociation from the TBC-DEG/TBCC platform.

# Materials and methods

## Recombinant expression and purification of tubulin cofactor complexes

Full length *S. cerevisiae* TBCC, TBCD, TBCE, and Arl2 cDNAs (also named Cin2, Cin1, Pac2, and Cin4, respectively) were amplified by PCR using oligonucleotides and inserted in two polycistronic bacterial expression vectors using isothermal assembly and confirmed by DNA sequencing. Each vector contains a single T7 promoter, individual ribosomal binding sites before each insert, and a single T7 terminator (*Tan et al., 2005*). To determine the accessibility of unique N- or C-termini of different TBC proteins, 6xHis or 6xHis-EGFP tags were inserted at either the 5′ or 3′ ends of TBCD, TBCE, or Arl2 cDNAs in different polycistronic expression vectors (as described 'Results' and shown in *Figure 2—figure supplement 1A,B*) and were tested for expression and purification, as described below. We determined the composition of TBC-DEG complexes purified from a TBCA, TBCB, TBCC, TBCD, TBCE, and Arl2 co-expression system using a nano LC-MS/MS approach, showing TBCD-E-Arl2 complexes (TBC-DEG) as described in *Table 1*. We focused on the study of TBC-DEG using two polycistronic vectors. We constructed modified TBC-DEG expression constructs, including a TBC-DEG-Arl2 Gln73Leu mutant, and EGFP inserted at the N-terminus of TBCE for further studies. TBCD, TBCE, and Arl2 deletion polycistronic

constructs (described in *Figure 2—figure supplement 1A*) were assembled through PCR by using inserts where cDNA sequences coding for either TBCD N-terminus (1–116 residues), TBCD C-terminus (866–1016 residues), TBCE N-terminal Cap-Gly domain (1–70 residues), and TBCE C-terminal ubiquitin domain (420–518 residues), Arl2 N-terminal (1–50 residues) and Arl2 C-terminus (90–125 residues) were deleted.

Recombinant TBC-DEG is purified as follows: polycistronic constructs were co-transformed into a bacterial expression strain at 37°C and then induced with 0.5 mM isothio-beta-glucopyranoside (IPTG) overnight at 20°C. Cells were pelleted and then lysed in 150 mM KCl, 50 mM HEPES, 1 mM MgCl$_2$, 3 mM β-mercaptoethanol, and 50 μM GTP with protease inhibitors including 1 mM PMSF, 1 μg/ml leupeptin, 20 μg/ml benzamidine, and 40 μg/ml aprotinin (RPI). The lysate was clarified by centrifugation at 18,000 rpm for 30 min at 4°C. Ni-NTA affinity (Macherey-Nagel, Bethlehem, PA, USA) was used to purify TBC-DEG complexes. NI-NTA purified TBC-DEG complexes were diluted with low salt buffer (100 mM KCl, 50 mM HEPES, 1 mM MgCl$_2$), bound to HiTrap SP FF (GE Healthcare, Pittsburg, PA, USA) anion exchange and then eluted with a five column volume gradient using high salt buffer (500 mM KCl, 50 mM HEPES, 1 mM MgCl$_2$). The TBC-DEG containing fractions were concentrated using Amicon concentrators and then loaded on a HiLoad 16/600 Superdex-200 gel filtration column (GE Healthcare). TBC-DEG was then used in subsequent studies as described below, without freezing.

Recombinant TBCC, its deletion and point mutants were expressed in bacteria. TBCC constructs were assembled using point mutagenesis and isothermal assembly, expressed in bacteria, and purified using the approach described above with few modifications. Briefly, bacterial cells overexpressing TBCC were resuspended in 50 mM MES, 100 mM KCl, and 3 mM β-mercaptoethanol, cells were lysed, and then the lysate was clarified by centrifugation as described above. TBCC was bound to Hitrap-SP FF and then eluted with a five column volumes gradient of 50 mM MES, 100 to 500 mM KCl, pH 6.0, and 3 mM β-mercaptoethanol. TBCC containing fractions were concentrated using Amicon concentrators and loaded on a Superdex 200 HiLoad 10/16 gel filtration column, analyzed by SDS-PAGE, and then frozen in liquid nitrogen.

## Biochemical assembly and analysis of tubulin cofactor-Arl2 αβ-tubulin complexes

Recombinant purified TBC-DEG (5–10 μmol) was diluted in 50 mM HEPES, 100 mM KCl, pH 7.0, and 3 mM β-mercaptoethanol including either GTP, GDP.ALF$_x$, or GTPγS nucleotide analogs, and then mixed with equimolar double-cycled porcine brain αβ-tubulin and/or TBCC. TBC-DEG, αβ-tubulin, and TBCC and their complexes were purified by size exclusion chromatography (SEC) using a Superdex 200 10/300 gel filtration column using an AKTA purifier (GE Healthcare) system, and 0.5 ml fractions were collected and analyzed using a Bis-Tris based XT criterion SDS-PAGE system (Bio-Rad, Hercules, CA, USA). The molecular masses of TBC-DEG, αβ-tubulin, TBCC, and their complexes were measured using SEC-MALS, proteins were separated on a WTC-03S5 size exclusion column (Wyatt Technologies, Santa Barbara, CA, USA), while UV absorbance (detected by Agilent 1100 Series HPLC), light scattering (Wyatt Technology miniDAWN TREOS), and refractive index (Wyatt Technology Optilab T-rEX) were measured and the concentration-weighted molecular weights of each peak were calculated using ASTRA V.6 software (Wyatt Technologies) (*Tarazona and Saiz, 2003*).

## Steady-state GTP hydrolysis measurement analysis

Steady-state GTP hydrolysis activity was measured using a malachite green free-phosphate detection assay as previously described (*Leonard et al., 2005*), with the following modifications: purified 10 μM recombinant TBC-DEG, αβ-tubulin, and TBCC were desalted using reaction buffer (50 mM HEPES, 100 mM KCl), combined in 96-well plates, in the presence of 0–800 μM GTP (Jena Biosciences, Jena, Germany), and incubated for 90 min at 30°C. The GTP hydrolysis reactions were quenched by the addition of 0.1 mM EDTA, followed by 1 mM malachite green. Phosphate-malachite green complex concentration was measured at 621 nm in a 96-well plate format using an Amersham plate reader (GE Healthcare). The phosphate concentration was determined using a 0–5 μM linear phosphate standard curve treated the same way as the reaction conditions. $K_m$ and $V_{max}$ were measured using a Michaelis–Menten curve fit. $V_{max}$ was used to calculate $k_{cat}$ values based on a 1 μmol TBC-DEG enzyme concentration and fit against a range of GTP substrate concentrations.

## Electron microscopy and single particle image analysis

Fresh SEC purified 0.5 mg/ml TBC-DEG-Q73L, NGFP-TBCE-DEG-Q73L, TBC-DEG-Q73L:αβ-tubulin, and TBC-DEG-Q73L:αβ-tubulin:TBCC complexes in 50 mM HEPES, 100 mM KCl, 0.1 mM GTP, and 3 mM β-mercaptoethanol were each incubated on carbon coated grids, briefly washed, and then stained with 1% uranyl formate. Electron microscopy was performed using a JEOL-2100 FF operating at 50,000 nominal magnification, and approximately 80–100 EM images were recorded on S0163 film (Kodak) for each condition, focusing mostly on areas of thick stain where particles are less likely to be flattened. Film images for each data set were scanned using a D8200 PrimeScan Heidelberg drum densitometer at 5.0 μm/pixel leading to 1 Å/pixel on the specimen. Images were normalized and binned two-fold (2 Å/pixel) using the EMAN2.1 software package (*Tang et al., 2007*). Roughly 18,000–20,000 globular cage-like individual TBC-DEG particles for each group were picked semi-automatically using e2boxer.py. Particle stacks were generated, then contrast transfer function (CTF) corrected with e2ctf.py using the phase flipping function. The image stacks were then subjected to iterative reference-free classification using e2refine2d.py generating 400 class averages, to remove roughly 10–30% of deformed, rare, or broken particle images. Additional rounds of classification were performed and the resulting 80-100 class averages show unique orientations suggesting a moderate degree of preferred orientations on the grids representing different, yet commonly observed views, as judged by their representation in the class averages (*Figure 4—figure supplement 1B*, *Figure 5—figure supplement 1B*, *Figure 7—figure supplement 1B*). For each data set, prominent class averages were then used to generate a starting model using a common lines strategy (*Tang et al., 2007*). These starting models were then filtered to 50 Å resolution and used in cycles of 3D projection matching and angular reconstitution using the SAMUEL program utilities running the SPIDER program (https://sites.google.com/site/maofuliao/samuel). Projection matching and angular reconstitution of the starting model were initiated at 30° increments and then decreased successively by 5° increments down to 5° (*Shaikh et al., 2008*) (*Figure 4—figure supplement 1C*, *Figure 5—figure supplement 1C*, *Figure 7—figure supplement 1C*). The resulting volume was filtered to 35 Å and the individual angular assignments were then further refined in multiple cycles in the program FREALIGN (*Grigorieff, 2007*; *Lyumkis et al., 2013*). Refinement convergence was determined from changes in phase residuals, angular assignment changes based on the program angplot_dp, and by comparing model projections to reference-free class averages with a global search using FREALIGN (*Figure 4—figure supplement 1D*, *Figure 5—figure supplement 1D*, *Figure 6—figure supplement 1D*). Fourier shell correlation (FSC) calculations from two half data sets indicate 25 Å resolution for all maps based on the 0.5 cutoffs (*Table 4*; *Figures 4—figure supplement 1E*, *Figure 5—figure supplement 1E*, *Figure 7—figure supplement 1E*). Final maps were aligned using the program XMIPP (*Sorzano et al., 2004*), were resolution truncated to nominal resolution based on FSC cutoffs, and were visualized using the program UCSF Chimera (*Pettersen et al., 2004*). The individual subunits were docked into the maps using two approaches that led to similar results. First, the maps were segmented using the segment map utility, and x-ray crystal structures for paralogs were fit using the fit-to-segment utility (*Pettersen et al., 2004*). Second, x-ray models were filtered to 24 Å resolution and then docked using the fit-in-map feature without segmentation, starting with the largest down to the smallest subunit, and each time cumulatively excluding regions of the map fit by the previous subunit. The floor and thumb regions (shown in *Figures 4E–H*, *Figure 5E–H*, and *Figure 7E–H* in pink) were fit with the Cse1 paralog x-ray structure (PDB-ID 1Z3H; *Cook et al., 2005a*), the TBCE bow region (shown in blue) was fit with the TLR4 LLR structure (PDB-ID 3FXI; *Park et al., 2009a*), its two globular end segments (shown in dark blue and cyan) were fit with the TBCB Cap-Gly structure (PDB-ID 4B6M; *Fleming et al., 2013a*) and the TBCB ubiquitin domain structure (PDB-ID 4B6W; *Fleming et al., 2013b*), respectively, the Arl2 pillar region (shown in orange) was fit with the human Arl2 structure (PDB-ID 1KSJ; *Hanzal-Bayer et al., 2002a*, *2002b*), αβ-tubulin dimer dual density (shown in red) was fit with the αβ-tubulin dimer (PDB-ID 1JFF; *Lowe et al., 2001*), the TBCC N-terminal domain segment (shown in yellow) was fit by the TBCC spectrin N-terminal domain structure (PDB ID 2L3L), and the wedge segment in the ternary complex map (*Figure 7E–H*, shown in green) was fit by the TBCC-C-terminal domain structure (determined here, PDB ID 5CYA). The final resolution truncated maps were deposited into the EMD database under accession numbers EMD-6393, EMD-6392, EMD-6391, and EMD-6390.

## X-ray crystallography and structure determination

Purified budding yeast TBCC was screened for crystallization in 96-well format using a mosquito crystallization robot (TTPlabtech, Oxford, UK), using a combination of home-made or commercial screens (Qiagen, Valencia, CA, USA). TBCC crystals formed in 0.1 M sodium citrate pH 5.6, 0.5 M ammonium sulfate, and 1.0 M lithium sulfate. The largest crystals were formed 1 week after micro-seeding in 0.1 M sodium citrate, 0.4 M ammonium sulfate, and 0.7 M lithium sulfate pH 5.2. Native TBCC crystals were soaked in mother liquor containing potassium hexacyanoplatinate, transferred to paratone-N oil, and then frozen in liquid nitrogen. TBCC diffraction data were collected from single crystals at the Stanford Synchrotron Radiation Laboratory (SSRL). The best TBCC crystals diffracted at 2.0 Å resolution in a tetragonal ($P\,4_3$) space group. Phase information was determined using platinum-substituted crystals using the multi-wavelength anomalous dispersion (MAD) approach with data collected in 10° wedge increments. TBCC diffraction data were indexed using the program MOSFILM in a $P\,4_3$ space group using the unit cell dimensions 70.03, 70.03, and 77.95 Å, and were scaled using the program SCALA (*Powell et al., 2013*). Phase information was determined by locating platinum atom positions using the program RESOLVE in the Phenix program suite (*Terwilliger and Berendzen, 1999*). The initial locations for platinum atom positions were determined, refined to a 0.69 figure of merit (FOM), and used to obtain initial TBCC density maps. TBCC density maps indicated two TBCC C-terminal domain molecules in the asymmetric unit; the TBCC N-terminal spectrin domain could not be identified in the density maps, suggesting it maybe disordered or underwent proteolysis during crystallization. A Matthew's coefficent calculation of the solvent content supports the idea that only the TBCC C-terminal domain is contained in the crystal rather than full length TBCC with a disordered N-terminus. A budding yeast TBCC C-terminal domain model was built starting at residue 100 and ending at residue 267 using the program COOT, and the resulting models were rigid-body refined using the Phenix program suite (*Emsley et al., 2010*; *Adams et al., 2010*).

To generate an Arl2-TBCC interface model, an RP2-Arl3 (PDB-ID 3BH7; *Veltel et al., 2008*) model was used as homology templates, where Arl2 (PDB-ID: 1KSJ; *Hanzal-Bayer et al., 2002b*) was aligned to Arl3, and the TBCC C-terminal domain, determined in this study (PDB-ID 5CYA), was aligned to RP2 using the Match-Maker structural analysis function in the program UCSF Chimera. The TBCC β-helix domain structure, homology models, and surface conservation images were generated using UCSF Chimera (*Pettersen et al., 2004*).

## Yeast genetic and cell biology analysis

Yeast manipulation, media, and transformation were performed by standard methods (*Amberg et al., 2005*). The Q73L substitution mutation was introduced into a CIN4 expression plasmid (pJM0230) by site-directed mutagenesis, creating a *cin4*-Q73L expression plasmid (pJM0231). Q73L was introduced into CIN4 at the endogenous locus using methods similar to those described in *Moore et al. (2009)*. All mutations were confirmed by sequencing the complete open reading frame.

Time=lapse images of cells expressing GFP-labeled microtubules (plasmid pSK1050, a gift from K Lee at the National Institutes of Health) were collected on a Nikon Ti-E microscope equipped with a 1.45 NA 100× CFI Plan Apo objective, piezo electric stage (Physik Instrumente, Auburn, MA), spinning disk confocal scanner unit (CSU10; Yokogawa), 488 nm laser (Agilent Technologies, Santa Clara, CA), and an EMCCD camera (iXon Ultra 897; Andor Technology, Belfast, UK) using NIS Elements software (Nikon). Living cells from asynchronous cultures grown to early log phase were suspended in non-fluorescent medium, mounted on a slab of 2% agarose, and sealed beneath a coverslip with paraffin wax. Z series of 13 images separated by 400 nm were collected. The number of aMTs was determined in the first Z series of each acquisition.

MT dynamics were analyzed by measuring aMT length at 4 or 5 s intervals for 10 min. This analysis was conducted in preanaphase cells, which typically exhibit one or two aMTs. Assembly and disassembly events were defined as continuous phases that produced a net change in aMT length of ≥0.5 μm and a coefficient of variation ≥0.85. Pause events were defined as lasting at least four data points (12 s) without significant change in aMT length. Catastrophes were defined as transitions from assembly or pause to disassembly. Catastrophe frequencies were determined for individual aMTs by dividing the number of catastrophe events by the total time spent in assembly and pause states. Rescues were defined as transitions from disassembly or pause to assembly. Rescue frequencies were determined for individual aMTs by dividing the number of rescue events by the total time for disassembly and pause states.

## Acknowledgements

We would like to thank Ruben Diaz-Avalos for collecting electron microscopy images using the Molecular Cellular biology electron microscopy facility, UC Davis. We thank Peter Dunten and staff at the Stanford Synchrotron Radiation Laboratory (SSRL) for assistance with TBCC x-ray diffraction. We thank Marjin Ford (University of Pittsburgh) for help and advice in TBCC structure determination. We thank Maofu Liao (Cell Biology, Harvard Medical School), Dimitry Lyumkis (Salk Institute) and Charles Sindular (Biochemistry and Molecular Biophysics, Yale University) for image processing advice and providing SAMUEL scripts. We thank Jodi Nunnari, Jonathan Scholey (Molecular Cellular Biology, University of California, Davis), Stephen C Harrison (Biological Chemistry and Molecular Pharmacology, Harvard Medical School), and Timothy Stearns (Biology, Stanford University) for discussion and support during various stages of this project. JAB is supported by NIH grant GM110283 and was previously supported by grant GM08429, JL acknowledges support from the NIH (GM047356), and JM from the NIH (GM092968). KDC acknowledges support from the Ludwig Institute for Cancer Research.

# Additional information

### Funding

| Funder | Grant reference | Author |
|---|---|---|
| National Institutes of Health (NIH) | GM110283, GM08429 | Jawdat Al-Bassam |
| National Institutes of Health (NIH) | GM092968 | Jeffrey K Moore |
| National Institutes of Health (NIH) | GM047356 | Julie Leary |

The funder had no role in study design, data collection and interpretation, or the decision to submit the work for publication.

### Author contributions

SN, SL, KDC, Conception and design, Acquisition of data, Analysis and interpretation of data, Drafting or revising the article; ES, JKM, Acquisition of data, Analysis and interpretation of data, Drafting or revising the article; WJ, Acquisition of data, Analysis and interpretation of data; JL, Conception and design, Analysis and interpretation of data, Contributed unpublished essential data or reagents; JA-B, Conception and design, Acquisition of data, Analysis and interpretation of data, Drafting or revising the article, Contributed unpublished essential data or reagents

# Additional files

### Major datasets

The following datasets were generated:

| Author(s) | Year | Dataset title | Dataset ID and/or URL | Database, license, and accessibility information |
|---|---|---|---|---|
| Nithianantham S, Le S, Seto E, Jia W, Leary J, Corbett KD, Moore JK, Al-Bassam J | 2015 | Crystal structure of Arl2 GTPase-activating protein tubulin cofactor C (TBCC) | http://www.rcsb.org/pdb/explore/explore.do?structureId=5CYA | Publicly available at the RCSB Protein Data Bank (Accession no: 5CYA). |
| Nithianantham S, Le S, Seto E, Jia W, Leary J, Corbett KD, Moore JK, Al-Bassam J | 2015 | TBC-DEG-Q73L | http://www.ebi.ac.uk/pdbe/entry/emdb/EMD-6390 | Publicly available at the Electron Microscopy Data Bank (accession no. EMD-6390). |
| Nithianantham S, Le S, Seto E, Jia W, Leary J, Corbett KD, Moore JK, Al-Bassam J | 2015 | TBC-DE(NGFP) | http://www.ebi.ac.uk/pdbe/entry/emdb/EMD-6391 | Publicly available at the Electron Microscopy Data Bank (accession no. EMD-6391). |

| Author(s) | Year | Dataset title | Dataset ID and/or URL | Database, license, and accessibility information |
|---|---|---|---|---|
| Nithianantham S, Le S, Seto E, Jia W, Leary J, Corbett KD, Moore JK, Al-Bassam J | 2015 | TBC-DEG-Q73L:tubulin | http://www.ebi.ac.uk/pdbe/entry/emdb/EMD-6392 | Publicly available at the Electron Microscopy Data Bank (accession no. EMD-6392). |
| Nithianantham S, Le S, Seto E, Jia W, Leary J, Corbett KD, Moore JK, Al-Bassam J | 2015 | TBC-DEG-Q73L:tubulin:TBCC | http://www.ebi.ac.uk/pdbe/entry/emdb/EMD-6393 | Publicly available at the Electron Microscopy Data Bank (accession no. EMD-6393). |

The following previously published datasets were used:

| Author(s) | Year | Dataset title | Dataset ID and/or URL | Database, license, and accessibility information |
|---|---|---|---|---|
| Cook A, Fernandez E, Lindner D, Ebert J, Schlenstedt G, Conti E | 2005 | The exportin Cse1 in its cargo-free, cytoplasmic state | http://www.rcsb.org/pdb/explore.do?structureId=1Z3H | Publicly available at the RCSB Protein Data Bank (Accession no: 1Z3H). |
| Fleming JR, Morgan RE, Fyfe PK, Kelly SM, Hunter WN | 2013 | Trypansoma brucei tubulin binding cofactor B CAP-Gly domain | http://www.rcsb.org/pdb/explore/explore.do?structureId=4B6M | Publicly available at the RCSB Protein Data Bank (Accession no: 4B6M). |
| Park BS, Song DH, Kim HM, Choi B-S, Lee H, Lee J-O | 2009 | Crystal structure of the human TLR4-human MD-2-E. coli LPS Ra complex | http://www.rcsb.org/pdb/explore/explore.do?structureId=3FXI | Publicly available at the RCSB Protein Data Bank (Accession no: 3FXI). |
| Hanzal-Bayer M, Renault L, Roversi P, Wittinghofer A, Hillig RC | 2002 | Complex of Arl2 and PDE delta, Crystal Form 2 (SeMet) | http://www.rcsb.org/pdb/explore/explore.do?structureId=1KSJ | Publicly available at the RCSB Protein Data Bank (Accession no: 1KSJ). |
| Fleming JR, Morgan RE, Fyfe PK, Kelly SM, Hunter WN | 2013 | Architecture of Trypanosoma brucei Tubulin-Binding cofactor B | http://www.rcsb.org/pdb/explore/explore.do?structureId=4B6W | Publicly available at the RCSB Protein Data Bank (Accession no: 4B6W). |
| Lowe J, Li H, Downing KH, Nogales E | 2001 | Refined structure of alpha-beta tubulin from zinc-induced sheets stabilized with taxol | http://www.rcsb.org/pdb/explore/explore.do?structureId=1JFF | Publicly available at the RCSB Protein Data Bank (Accession no: 1JFF). |

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
