## [Decision Letter]

Thank you for submitting your work entitled “Tubulin Cofactors and Arl2 are
Cage-like Chaperones that regulate soluble αβ-Tubulin pools for
Microtubule Dynamics” for peer review at *eLife*. Your submission
has been favorably evaluated by Randy Schekman (Senior Editor) and three reviewers, one
of whom is a member of our Board of Reviewing Editors.

The reviewers have discussed the reviews with one another, and the Reviewing Editor has
drafted this decision to help you prepare a revised submission.

Summary:

This manuscript describes ambitious work to reconstitute and characterise
multi-component tubulin chaperone complexes. This is an important but poorly understood
facet of tubulin homeostasis, and the authors have expended considerable effort to study
these complexes using a range of methods. The authors have purified and characterised
the complex containing the yeast tubulin cofactors D and E together with the GTPase Arl2
and showed that they form a stable heterotrimer, which they term TBC-DEG. Using negative
stain EM data, the authors investigated the structure of the TBC-DEG complex. They
showed that the binding of tubulin dimer and the tubulin cofactor C activates GTP
hydrolysis, and provided evidence that the ability of Arl2 to hydrolyse GTP is
functionally important. Finally, the authors explored the activity of the chaperone
towards tubulin using microscopy-based microtubule dynamics assays.

Several parts of the work presented – e.g. the biochemical characterisation of
the chaperone components and the yeast TBCC C-terminal domain crystal structure –
shed new light on aspects of the chaperone mechanism. However, there are some
significant concerns, particularly about the 3D reconstruction using negative stain EM
data, which would need to be addressed. The data on microtubule dynamics appear
difficult to interpret, and the authors should very seriously consider saving them for
another manuscript, where the effect of the complex on microtubule growth is
investigated more thoroughly.

Essential revisions:

1) Table 1 indicates that TBCC peptide coverage
is 0.9%. Is this a typo? If not, what is the protein that has been studied?

2) In the first paragraph of the subsection “Tubulin cofactors D, E and Arl2
GTPase form a stable heterotrimeric TBC-DEG chaperone” the author states that
‘Monomeric TBCD, TBCE and Arl2 subunits were not observed in vitro at any
concentration and the TBC-DEG complex behaves as a single biochemical entity’.
Could it be that the subunits were not recovered on the column due to their tendency to
precipitate?

3) 3D reconstruction using negative stain EM data is a significant proportion of the
assembled evidence. Although intrinsically limited in achievable resolution, use of
negative stain is legitimate with such small and apparently aggregation-prone complexes.
However, as outlined below, the analysis of these EM data does not appear to be
sufficiently robust and needs to be improved, in order to ensure that all potential
errors and/or ambiguities are eliminated.

The authors have taken a systematic approach to reconstructing a series of multi-subunit
complexes applying standard methodologies. However, for these standard methods to work,
several conditions must be met:

A) Individual particles and class averages must be centred: based on the views in Figure 3—figure supplement 1, this does
not appear to be the case – e.g. top row, second from right, fourth row, first on
left;

B) Classes must be homogeneous – however multiple class averages show a lack of
crispness that is suggestive of member heterogeneity – e.g. second row 9 along;
third row 7 along. This problem is particularly acute in the larger complexes shown in
Figure 6—figure supplement 1 but
also in Figure 4—figure supplement 1;
note also there are no stars in these figures, as suggested by the legend.

When these conditions are not met, structures cannot be considered to be reliable.
Unreliability is further supported by the lack of match between class averages and
re-projections e.g. in Figure 3—figure supplement 1, Figure 4—figure supplement 1 and Figure 6—figure supplement 1.

The segmentation imposed on such lower resolution reconstructions is essential for
interpretation, but appears a bit arbitrary. Putting aside concerns about the structures
themselves, the evidence presented for subdomain assignment is not currently justified.
First, the high cross-correlation (CC) values from the docking calculations are not by
themselves evidence of a unique assignment – for example, what was the CC when
the HEAT repeats were fit in the pink density rather than the blue density? The lack of
connectivity between the TBCE LRR and Ubiquitin is particularly unsatisfying –
and the CAPGly and LRR are separated by ∼100A in the final +tub
+TBCC reconstruction, which seems unlikely. Second, the data localising the GFP
density in two different tagged versions of the TBC-DEG complex (Figure 3–figure
supplement 2) are unconvincing because: a) the density for the GFP in the class averages
is not always visible and b) it is not obvious that the same projections are being
compared {+/-}GFP.

4) The experiments with dynamic microtubules represent the weakest part of the story.
The authors show that in the presence of 6μM tubulin, nanomolar concentrations of
TBC-DEG moderately reduce microtubule growth rate and promote rescues in a
concentration-dependent manner. At concentrations above 20 nM, the effect on rescues is
reduced and a bi-modal distribution of catastrophe frequencies is observed. When the
concentration is increased further, microtubule pausing is observed. When the TBC-DEG
complex version containing the GTP-locked Arl2-Q73L is used, very slow growth or even
microtubule pausing is observed. The authors interpret the experiments in terms of the
effects on the tubulin dimer (dimer re-activation, sequestration or decay, dependent on
concentration, although there is very little evidence to make a justified choice between
these different possibilities). However, it is well known that when tubulin
concentration is increased, microtubules normally should grow faster. Slow growth,
pausing and rescues – very prominent phenotypes observed by the authors –
are all typically associated with proteins acting on microtubule ends or on the
microtubule lattice. The authors suggest that their complexes do not bind to microtubule
ends, but the data shown (Figure 7) are clearly
not of sufficient quality to make a strong conclusion about this: TBC-DEG-Q73L strongly
binds to the GMPCPP seeds and the overall background is high. Further, it is not shown
whether the ReAsH labelled complex has the same activity as the unlabelled one. The only
way to exclude the direct effects of the cofactors on the microtubules would be to
pre-incubate the tubulin with the cofactor for different times, remove the cofactor and
perform microtubule dynamics assays. Based on these considerations, the authors are
strongly encouraged to remove this section and save the results for a subsequent study,
where the effects of the complex on microtubule dynamics are researched properly.

5) It remains difficult to understand the effect and the importance of GTP hydrolysis by
Arl2 for the TBC-DEG cycle. While no additional experiments seem to be necessary, the
authors should make a much better effort to describe their interpretation of the results
discussed in the section “Sequential binding of tubulin and TBCC activates
maximal GTP hydrolysis in TBC-DEG chaperones” and in the Discussion (second
paragraph). What do the authors mean when they say “Arl2 GTPase state likely
controls the TBC-DEG chaperone state” – to which part of the chaperone
cycle does this refer? For many small GTPases, locking them a GTP state makes them
constitutively active, and indeed Q73L-Arl2 promotes formation of the complex of TBC-DEG
with tubulin and TBCC. Is tubulin recycling blocked in this case? If so, the complex
containing Q73L-Arl2 will act as a stable tubulin sequestering agent.

[Editors' note: further revisions were requested prior to acceptance, as
described below.]

Thank you for resubmitting your work entitled “Tubulin Cofactors and Arl2 are
Cage-like Chaperones that regulate the soluble αβ-Tubulin pool for
Microtubule Dynamics” for further consideration at *eLife*. Your
revised article has been favorably evaluated by Randy Schekman (Senior Editor) and a
Reviewing Editor. The manuscript has been improved but there are some remaining issues
that need to be addressed before acceptance, as outlined below:

1) Figure 4—figure supplement 1 seems to
contain colored asterisks and some other material hidden behind panel B. This probably
reflects inaccurate figure preparation and must be removed.

2) In Figure 4—figure supplement 1
(panels B and D), some of the particles are still not centered within the box. This is
most apparent with donut-shaped averages where dark center of particle should be in
middle of box e.g. top row, 3rd from left: original critique of these data still
applies.

3) Please compare your work to the paper by Serna et al. in J Cell Science (PMID:
25908846).

---

## [Author Response]

*Several parts of the work presented – e.g. the biochemical
characterisation of the chaperone components and the yeast TBCC C-terminal domain
crystal structure – shed new light on aspects of the chaperone mechanism.
However, there are some significant concerns, particularly about the 3D
reconstruction using negative stain EM data, which would need to be addressed. The
data on microtubule dynamics appear difficult to interpret, and the authors should
very seriously consider saving them for another manuscript, where the effect of the
complex on microtubule growth is investigated more thoroughly*.

We thank the editor and the reviewers for their enthusiasm for our manuscript, and for
the thoughtful advice in revising the manuscript. We have fully revised the manuscript
in order to address all comments. Major areas of revision include:

1) Supporting negative-stain electron microscopy analyses. We present revised
multivariant statistical analysis, and comparisons of the model projections to image
class averages for each of the structures. We also provide a parallel molecular docking
approach independent of segmentation to support our docking of various molecular models
into the refined EM density.

2) As suggested by the reviewers, we have removed our TIRF-based experiments on the
effects of the tubulin cofactors on microtubules dynamics. We are further revising these
studies and we will be publishing those studies in separate manuscript elsewhere.

3) We have significantly revised our results and discussion of GTP hydrolysis studies,
to make both the results and our interpretations clearer.

4) We have revised and improved the manuscript throughout to clarify the language and
improve cohesion of the manuscript.

*Essential revisions*:

*1)*
Table 1
*indicates that TBCC peptide coverage is 0.9%. Is this a typo? If not, what is
the protein that has been studied?*

The mass-spectrometry analysis of the tubulin cofactor complex was carried out from a
bacterial co-expression of TBCA, TBCB, TBCC, TBCD, TBCE and Arl2 with an individual
his-tag on TBCD. The sample analyzed included purified materials following nickel
affinity chromatography. The low coverage of 9% for TBCC is due to the low affinity of
TBCC to the TBC-DEG complex, which is successively lost in the later purification steps.
These have been further explained in the manuscript.

2) In the first paragraph of the subsection “Tubulin cofactors D, E and
Arl2 GTPase form a stable heterotrimeric TBC-DEG chaperone” the author states
that ‘Monomeric TBCD, TBCE and Arl2 subunits were not observed in vitro at any
concentration and the TBC-DEG complex behaves as a single biochemical entity’.
Could it be that the subunits were not recovered on the column due to their tendency
to precipitate?

We agree with the reviewers that it is likely that any individual TBC-DEG subunits
precipitate if not integrated into the stable complex. This is supported by our finding
that the individually expressed TBCD, TBCE and Arl2 subunits are insoluble.

*3) 3D reconstruction using negative stain EM data is a significant proportion of
the assembled evidence. Although intrinsically limited in achievable resolution, use
of negative stain is legitimate with such small and apparently aggregation-prone
complexes. However, as outlined below, the analysis of these EM data does not appear
to be sufficiently robust and needs to be improved, in order to ensure that all
potential errors and/or ambiguities are eliminated*.

We appreciate the reviewers’ suggestions on revising the electron microscopy
analyses. These suggestions have improved our manuscript and the revised manuscript
presents these changes as described below.

The authors have taken a systematic approach to reconstructing a series of
multi-subunit complexes applying standard methodologies. However, for these standard
methods to work, several conditions must be met:

*A) Individual particles and class averages must be centred: based on the views
in*
Figure 3—figure supplement 1*, this does not appear to be the case – e.g. top
row, second from right, fourth row, first on left*;

We have revised our multi-variant statistical analysis (MSA) particle image
classification for the structures described. These data now show centered class averages
in Figure 4—figure supplement 1, Figure 5—figure supplement 1 and Figure 7—figure supplement 1.

*B) Classes must be homogeneous – however multiple class averages show a
lack of crispness that is suggestive of member heterogeneity – e.g. second row
9 along; third row 7 along. This problem is particularly acute in the larger
complexes shown in*
Figure 6—figure supplement 1
*but also in*
Figure 4—figure supplement 1*; note also there are no stars in these figures, as
suggested by the legend*.

We have revised our multivariant statistical analysis (MSA) particle image
classification to calculate fewer class averages to encompass more images per class
average. Our revised analysis show crisp class averages, as expected, and show more of
the features of each view of the complex. These are now shown in Figure 4—figure supplement 1, Figure 5—figure supplement 1 and Figure 7—figure supplement 1.

*When these conditions are not met, structures cannot be considered to be
reliable. Unreliability is further supported by the lack of match between class
averages and re-projections e.g. in*
Figure 3—figure supplement 1*,*
Figure 4—figure supplement 1
*and*
Figure 6—figure supplement 1.

We present a revised projection-matching using the approach described by [37] (Science*,* 342
(6165) 1484-1490), where global angular search was carried out using the MSA class
averages to identify and refine the matching model projections to each class average.
Our revised analyses show the MSA class averages closely match projections from models
presented for each structure. These are presented in Figure 4—figure supplement 1, Figure 5—figure supplement 1 and Figure 7—figure supplement 1.

The segmentation imposed on such lower resolution reconstructions is essential
for interpretation, but appears a bit arbitrary. Putting aside concerns about the
structures themselves, the evidence presented for subdomain assignment is not
currently justified. First, the high cross-correlation (CC) values from the docking
calculations are not by themselves evidence of a unique assignment – for
example, what was the CC when the HEAT repeats were fit in the pink density rather
than the blue density?

We completely agree with the reviewers that the segmentation of low-resolution negative
stain electron microscopy maps involves a degree of arbitrariness. However, our maps
provide enough shape information that molecular models can be fit to unique positions
within these maps even without using segmentation. There is a moderate to high sequence
homology between TBCD and TBCE-LLR (40% identical) to the paralog structures being used
for the fitting. We used an independent approach for fitting, presented in Figure 4—figure supplement 3.
Low-resolution models for TBCD, TBCE paralogs and Arl2, generated from high-resolution
structures or homology models, were cumulatively fit into the TBC-DEG map. The paralog
structure for TBCD, the largest subunit, has a distinctive “ring with rod”
shape, which can be fit fairly unambiguously. Once TBCD is positioned and that region of
the map is excluded, the TBCE-LLR can only be fit reasonably into the bow density with
two globular densities, attached both ends of the bow, likely to be TBCE Cap-Gly and
C-terminal ubiquitin domains. Finally, the more globular Arl2 matches the size of the
remaining pillar density.

*The lack of connectivity between the TBCE LRR and Ubiquitin is particularly
unsatisfying – and the CAPGly and LRR are separated by ∼100A in the
final +tub +TBCC reconstruction, which seems unlikely*.

The lack of connectivity between the TBCE-LLR and Ubiquitin domain is due to our choice
of contour-threshold chosen to present the TBC-DEG map or potential density defects
caused by negative stain. We describe these details in the revised manuscript in the
subsections “The TBC-DEG complex is a cage-like chaperone with a hollow central
core” and “Electron microscopy and Single particle image
analysis”.

We have reevaluated the positioning of the Cap-Gly domain in the TBC-DEG
Q73L:αβ-tubulin:TBCC map based on the reviewer’s comment. We agree
that the Cap-Gly domain placement was likely incorrect in the previous model. We have
generated a revised model for the TBC-DEG-αβ-tubulin: TBCC complex in
comparison to its previous positions in the TBC-DEG and TBC-DEG:αβ-tubulin
maps. We believe the revised model takes into account the positioning of TBCE-terminal
domains more accurately. We have further described these details in the revised
manuscript in the subsections “TBCC β-helix wedge interfaces with Arl2 and
αβ-tubulin dimer in the TBC-DEG chaperone” and “Electron
microscopy and Single particle image analysis”.

However, higher resolution structures will be required to further understand the
conformations of individual domains in each of the different states.

*Second, the data localising the GFP density in two different tagged versions of
the TBC-DEG complex (Figure 3–figure supplement 2) are unconvincing because:
a) the density for the GFP in the class averages is not always visible and b) it is
not obvious that the same projections are being compared
{*+/-*}GFP*.

We have revised the MSA particle image classification of GFP-fusion TBC-DEG complexes,
as described above for the native complex. We have used projection matching to compare
the TBC-DEG and TBC-DE(GFP)G projections and have identified the correct projections
more accurately. The GFP density appears clearly in most revised projections.

*4) The experiments with dynamic microtubules represent the weakest part of the
story. The authors show that in the presence of 6μM tubulin, nanomolar
concentrations of TBC-DEG moderately reduce microtubule growth rate and promote
rescues in a concentration-dependent manner. At concentrations above 20 nM, the
effect on rescues is reduced and a bi-modal distribution of catastrophe frequencies
is observed. When the concentration is increased further, microtubule pausing is
observed. When the TBC-DEG complex version containing the GTP-locked Arl2-Q73L is
used, very slow growth or even microtubule pausing is observed. The authors interpret
the experiments in terms of the effects on the tubulin dimer (dimer re-activation,
sequestration or decay, dependent on concentration, although there is very little
evidence to make a justified choice between these different possibilities). However,
it is well known that when tubulin concentration is increased, microtubules normally
should grow faster. Slow growth, pausing and rescues – very prominent
phenotypes observed by the authors – are all typically associated with
proteins acting on microtubule ends or on the microtubule lattice. The authors
suggest that their complexes do not bind to microtubule ends, but the data shown
(*Figure 7*)
are clearly not of sufficient quality to make a strong conclusion about this:
TBC-DEG-Q73L strongly binds to the GMPCPP seeds and the overall background is high.
Further, it is not shown whether the ReAsH labelled complex has the same activity as
the unlabelled one. The only way to exclude the direct effects of the cofactors on
the microtubules would be to pre-incubate the tubulin with the cofactor for different
times, remove the cofactor and perform microtubule dynamics assays. Based on these
considerations, the authors are strongly encouraged to remove this section and save
the results for a subsequent study, where the effects of the complex on microtubule
dynamics are researched properly*.

We agree with the reviewers that the regulatory effects of tubulin cofactor chaperone on
the soluble tubulin pools and their dynamic polymerization into microtubules are
complex. The defects in GTP hydrolysis observed in the Arl2 Q73L mutant, however, lead
to pausing of microtubule dynamics, that we observed both in vivo and in vitro.

As the reviewers requested, we have removed these data from the manuscript.

*5) It remains difficult to understand the effect and the importance of GTP
hydrolysis by Arl2 for the TBC-DEG cycle. While no additional experiments seem to be
necessary, the authors should make a much better effort to describe their
interpretation of the results discussed in the section “Sequential binding of
tubulin and TBCC activates maximal GTP hydrolysis in TBC-DEG chaperones” and
in the Discussion (second paragraph). What do the authors mean when they say
“Arl2 GTPase state likely controls the TBC-DEG chaperone state”
– to which part of the chaperone cycle does this refer? For many small
GTPases, locking them a GTP state makes them constitutively active, and indeed
Q73L-Arl2 promotes formation of the complex of TBC-DEG with tubulin and TBCC. Is
tubulin recycling blocked in this case? If so, the complex containing Q73L-Arl2 will
act as a stable tubulin sequestering agent*.

We have revised the text in the Results and Discussion sections to further clarify the
role GTP-hydrolysis to address all the points described above regarding the mechanism of
the tubulin cofactors/Arl2 chaperone system.

[Editors' note: further revisions were requested prior to acceptance, as
described below.]

*1)*
Figure 4—figure supplement 1
*seems to contain colored asterisks and some other material hidden behind panel
B. This probably reflects inaccurate figure preparation and must be
removed*.

We thank the reviewer for noticing this error. We have removed these asterisks in the
revised Figure 4—figure supplement 1.

*2) In*
Figure 4—figure supplement 1
*(panels B and D), some of the particles are still not centered within the box.
This is most apparent with donut-shaped averages where dark center of particle should
be in middle of box e.g. top row, 3rd from left: original critique of these data
still applies*.

The reviewer is correct that the class average images in Figure 4—figure supplement 1 are slightly off-center.
Indeed, in these images all of the class averages are 1-2 pixels to the left of center.
This is an image masking/display artifact, and has no effect on the quality of the
resulting reconstruction or on the quality of our multivariant statistical analyses
classification. Each individual image used for the electron microscopy reconstructions
and validation of model projections are independently centered prior to their inclusion
in projection matching or angular refinement algorithms.

*3) Please compare your work to the paper by Serna et al. in J Cell Science
(PMID: 25908846)*.

We agree that the study above is important, and have added the reference in the Results
section of the manuscript where appropriate. Our conclusions, however, potentially
differ from the conclusions of Serna et al., and we do not observe a TBCB-TBCE complex
in our work. Our approach to co-express the TBC proteins with different tags did not
lead to TBCB interacting with TBCE under any condition. In addition, we never observed
isolated TBCE and thus our data indicate that recombinant TBCE is likely insoluble
without TBCD and Arl2.

It is possible that the TBCE protein described in the Serna et al. manuscript is
metastable. Alternatively, TBCD and Arl2 may have co-purified from insect cells where
the purification of TBCE was carried out. The lack of mass-spectrometry data in this
manuscript prevents us from comparing the TBCE preparation to that we present.

Despite these outstanding issues, we think it is significant that the overall shape and
organization of the low resolution TBCE reconstruction presented by Serna et al.
suggests a similar organization to the region we assign as TBCE in our map. The crescent
shape that we assign to the TBCE LLR domain with its two globular densities supports our
conclusions regarding TBCE organization, with terminal Cap-Gly and ubiquitin
domains.